# High-accuracy and dimension-free sampling with diffusions

**Khashayar Gatmiry** [1][2]   **Sitan Chen** [3]   **Adil Salim** [4][2]

## Abstract

Diffusion models have shown remarkable empirical success in sampling from rich multi-modal distributions. Their inference relies on numerically solving a certain differential equation. This differential equation cannot be solved in closed form, and its resolution via discretization typically requires many small iterations to produce *high-quality* samples. More precisely, prior works have shown that the iteration complexity of discretization methods for diffusion models scales polynomially in the ambient dimension and the inverse accuracy $1/\varepsilon$. In this work, we propose a new solver for diffusion models relying on a subtle interplay between low-degree approximation and the collocation method (Lee et al., 2018), and we prove that its iteration complexity scales *polylogarithmically* in $1/\varepsilon$, yielding the first "high-accuracy" guarantee for a diffusion-based sampler that only uses (approximate) access to the scores of the data distribution. In addition, our bound does not depend explicitly on the ambient dimension; more precisely, the dimension affects the complexity of our solver only through the *effective radius* of the support of the target distribution.

## 1. Introduction

Diffusion models (Sohl-Dickstein et al., 2015; Song & Ermon, 2019; Ho et al., 2020; Dhariwal & Nichol, 2021; Song et al., 2021b;a; Vahdat et al., 2021) are the dominant paradigm in image generation, among other modalities. They sample from high-dimensional distributions by numerically simulating a *reverse process* driven by a certain differential equation with drift learned from training data. The reverse process is meant to undo some noise process, and by simulating the reverse process sufficiently accurately, one can generate fresh samples out of pure noise.

The empirical success of this method has spurred a flurry of theoretical work in recent years to understand the mechanisms by which diffusion models are able to easily sample from rich multimodal distributions. These works draw upon tools from the extensive literature on log-concave sampling and arrive at a remarkable conclusion: diffusion models can efficiently sample from *any* distribution in high dimensions, even highly non-log-concave ones, provided one has access to a sufficiently accurate estimate of the score of the distribution along a noise process (see Section 1.2 for an overview of this literature).

The earliest such results showed that for any smooth distribution with bounded second moment, one can sample to error $\varepsilon$ in total variation distance in $\mathrm{poly}(d, 1/\varepsilon)$ iterations, given sufficiently accurate score estimation (Lee et al., 2023; Chen et al., 2023b;a; Benton et al., 2024; Conforti et al., 2025). These results focus on samplers that simulate the reverse process in discrete time. The discretization introduces bias, which necessitates taking the discretization steps sufficiently small – i.e. of size $\mathrm{poly}(\varepsilon, 1/d)$ – to ensure the trajectory of the sampler remains sufficiently close to that of the continuous-time reverse process.

While these discretization bounds have been refined significantly by recent work (see Section 1.2), what remains poorly understood is the dependence on $\varepsilon$. In the log-concave sampling literature, there is a well-studied taxonomy along this axis: there are (1) "low-accuracy" methods like Langevin Monte Carlo that get iteration complexity scaling polynomially in $1/\varepsilon$, and (2) "high-accuracy" methods which correct for discretization bias, e.g., via Metropolis adjustment, and get iteration complexity *polylogarithmic* in $1/\varepsilon$.

For diffusion models, guarantees of the second kind remain largely unexplored. We therefore ask:

> *Are there samplers for diffusion modeling that achieve $O(\mathrm{polylog}\,1/\varepsilon)$ iteration complexity?*

### 1.1. Our contributions

Here we answer this in the affirmative. We will focus on the following class of target distributions $q$:

**Assumption 1** (Bounded plus noise)**.** *Let $R, \sigma > 0$. There exists a distribution $q_{\mathsf{pre}}$ supported on the origin-centered ball of radius $R$ in $\mathbb{R}^d$ for which $q = q_{\mathsf{pre}} \star \mathcal{N}(0, \sigma^2 I)$.*

---

[1]UC Berkeley [2]Part of this work was done at Microsoft Research [3]Harvard University [4]Stealth. Correspondence to: Adil Salim <asalim.math@gmail.com>.

*Proceedings of the 43$^{rd}$ International Conference on Machine Learning*, Seoul, South Korea. PMLR 306, 2026. Copyright 2026 by the author(s).

In other words, we consider sampling from a distribution $q$ which is the convolution of a compactly supported distribution with Gaussian noise. Our bounds will scale polynomially in $R/\sigma$. A natural example of a distribution satisfying Assumption 1 is a *mixture of isotropic Gaussians*; more generally, as observed by Chen et al. (2023b), distributions satisfying Assumption 1 naturally model what diffusion models in practice try to sample from via *early stopping*. Positivity of $\sigma$ also makes it possible to prove convergence guarantees in strong divergence-based metrics like total variation distance.

In this paper, we consider sampling from $q$ by approximately simulating the reverse process. We first show a key structural property: the high-order time derivatives of the score function along the reverse process are bounded, implying that the score function can be pointwise-approximated by a low-degree polynomial in time. The statement is technical and we defer it to the supplement.

Leveraging this result, we show how to adapt a certain specialized ODE solver of Lee et al. (2018) to simulate steps along the reverse process, and argue that our simulation remains close to the trajectory of the reverse process at all times, see Algorithm 2. We thus prove that Algorithm 2 can efficiently sample from $q$, yielding the following main guarantee:

**Theorem 1.1** (Informal, see Corollary 3.9). *Let $\varepsilon, R, \sigma > 0$. Suppose $q$ is a distribution satisfying Assumption 1 for parameters $R, \sigma$. Given access to Lipschitz score estimates for which the estimation error is $L^2$-bounded and has subexponential tails, there is a diffusion-based algorithm (see Algorithm 2) which outputs samples from a distribution $\varepsilon$ close to $q$ in total variation (TV) distance in $\tilde{O}((R/\sigma)^2 \cdot \operatorname{poly} \log 1/\varepsilon)$ iterations.*

To the best of our knowledge, this is the first diffusion-based sampler to achieve polylogarithmic, rather than polynomial, dependence on $1/\varepsilon$, while only assuming access to a sufficiently accurate score. While our result requires a stronger assumption on the distribution of score error than prior work, we note that the analysis in prior work incurs polynomial dependence on $1/\varepsilon$ *even assuming perfect score estimation*.

Another appealing feature of our result is that the iteration complexity of our algorithm is *independent* of the ambient dimension $d$, instead depending on the radius $R$ in Assumption 1. While in general $R$ can scale with $d$, the Gaussian mixture setting offers a natural setting where $R$ can be much smaller than $d$. For example, in the theory of distribution learning, the most challenging regime is precisely when the component centers are at distance $\Theta(\sqrt{\log k})$ from each other (Hopkins & Li, 2018; Kothari et al., 2018; Diakonikolas et al., 2018; Liu & Li, 2022). This corresponds to the case of $R = \Theta(\sqrt{\log k})$ and $\sigma = 1$, for which our sampler achieves iteration complexity scaling only polylogarithmi-

cally in $k$ and $1/\varepsilon$, whereas existing diffusion-based methods would scale polynomially in $d$ and $1/\varepsilon$ for this example (see Section 1.2 for discussion of a recent result (Li et al., 2025a) that obtains a dimension-free rate in the Gaussian mixture setting).

## 1.2. Related work

We review the extensive line of recent works on diffusion model convergence bounds in Appendix A, focusing here on the threads most relevant to our result.

**Dependence on $\varepsilon$.** First, mirroring the popularization of higher-order solvers for diffusion sampling in practice (Lu et al., 2022; 2025; Zhao et al., 2023), a number of works have sought to improve the $\varepsilon$ dependence for diffusion-based sampling by appealing to ideas from higher-order numerical solvers, e.g., Li et al. (2024a); Wu et al. (2024); Li et al. (2024a) which achieved *fixed* polynomial acceleration in the $\varepsilon$ dependence under relatively minimal assumptions.

The recent works of Huang et al. (2025a;b) and Li et al. (2025b) used higher-order solvers to achieve *arbitrary* polynomial acceleration, i.e. a rate scaling like $O(d^{1+O(1/K)}/\varepsilon^{1/K})$ for arbitrarily large constant $K$. We note however while this might suggest that one could take $K = \tilde{\Theta}(\log(1/\varepsilon))$ and achieve similar polylogarithmic scaling as our work, the analysis in these works implicitly assumes that the number of iterations is at least exponential in $K$ (see e.g., the Discussion in Li et al. (2025b)), so the acceleration they obtain is at best polynomial rather than exponential like in our work. In addition, the assumptions in these works are incomparable to ours. For instance, Huang et al. (2025a;b) assume at least *second-order* smoothness of the score estimates; Li et al. (2025b) does not require any smoothness of the score estimate, but it assumes its Jacobian of the score estimate is close to that of the true score. In contrast, we need to assume *first-order* smoothness of the score estimate (which is justified because the true score is Lipschitz) and sub-exponential score error. Finally, we note that these three works and ours all assume the data distribution has compact support.

To our knowledge, the only prior work which truly achieved exponentially improved $\varepsilon$ dependence is Huang et al. (2024a). However, that work applies in a stronger access model than is commonly studied in this literature, as they assume approximate query access to log density ratios between arbitrary points, in addition to the scores, to implement a Metropolis-Hastings filter. We also note the independent and concurrent work of Wainwright (2025) which decomposes the reverse process into sampling from a sequences of posteriors which are strongly log-concave, and notes that by using off-the-shelf high-accuracy guarantees for strongly log-concave sampling, one can achieve a loga-

rithmic dependence on $\varepsilon$ for diffusion-based sampling. However, these off-the-shelf routines require knowledge not just of the scores, but of the log density of the data distribution convolved with noise, similar to Huang et al. (2024a).

**Dimension-free bounds.** As for our dependence on $R/\sigma$ instead of $d$, there is a relevant line of work (Li & Yan, 2024a; Liang et al., 2025; Huang et al., 2024b; Potaptchik et al., 2024) on sampling from distributions with low intrinsic dimension. They show that for any distribution whose intrinsic dimension, which they quantify in terms of covering number of the support denoted $k$, DDPM can sample with iteration complexity $O(k^4/\varepsilon^2)$. While this provides another example of a diffusion-based sampler whose rate adapts to the underlying geometry of the distribution, it is incomparable to our result: there can be supports which are high-dimensional but bounded in radius, and vice versa. On the other hand, Li et al. (2025a) considered the special case of mixtures of isotropic Gaussians and showed, using very different techniques from those of the present work, that diffusion models can sample from these distributions in $T = \tilde{O}(\text{polylog}(k)/\varepsilon)$ iterations, provided that the means of the Gaussians have norm at most $T^C$ for arbitrarily large absolute constant $C$. We remark that this result is incomparable to ours. We obtain polylogarithmic dependence on $1/\varepsilon$ and our guarantee applies to a wide class of distributions beyond just the special case of mixtures of isotropic Gaussians. On the other hand, the result of Li et al. (2025a) has superior scaling in other parameters for Gaussian mixtures: whereas we have a polynomial dependence on the maximum norm of any center in the mixture, they have an *arbitrarily small* dependence on the radius.

**Collocation for diffusions.** Finally, we note that the collocation method (see Section 2.3) has been studied in the context of diffusions, but primarily as a way to *parallelize* the steps of the sampler (Anari et al., 2023; Gupta et al., 2024a; Chen et al., 2024a), but not using low-degree polynomial approximation. Those works obtain polylogarithmic *round complexity*, but the *total work* done is still polynomial in the target accuracy. In contrast, the key technical ingredient in our work is to establish low-degree approximability of the score by controlling its time derivatives, which is similar in spirit to Section 3 of Stéphanovitch (2025), even if our results are incomparable. This key ingredient allows us to establish that the implementation of collocation by Lee et al. (2018) via low-degree polynomials over a small time window of the probability flow ODE converges at an *exponential* rate, even in a sequential setting.

**High-accuracy log-concave sampling.** The literature on high-accuracy samplers, which is traditionally centered around log-concave distributions and more generally distributions which satisfy a functional inequality, is too extensive to do justice to here, and we refer to Chapter 7 of Chewi (2023) for a detailed overview. We briefly overview this in Appendix A. Our techniques do not draw upon this literature but are instead based on the work of Lee et al. (2018), which devised the general framework of collocation via low-degree approximation that we employ. Motivated by sampling problems connected to logistic regression, they applied their method to obtain a high-accuracy sampler for densities of the form $q \propto e^{-f}$, where $f(x) = \sum_i \phi_i(\langle a_i, x \rangle) + \lambda \|x\|^2$.

### 1.3. Roadmap

In Section 2, we provide technical preliminaries and give a description of our sampler. We also provide intuition for how this sampler can achieve the high-accuracy guarantee of Theorem 1.1. In Section 3, we outline the proof of our main structural result showing that the drift of the reverse process is well-approximated by a low-degree polynomial in time, as well as the main steps for completing the proof of Theorem 1.1. We defer the full proof to the Appendix.

## 2. Preliminaries and our algorithm

**Notation.** Let $\gamma^d$ denote the $d$-dimensional standard Gaussian distribution. Given interval $I \subseteq \mathbb{R}^d_{\geq 0}$, let $\mathcal{C}(I, \mathbb{R}^d)$ denote the space of continuous maps $I \to \mathbb{R}^d$. For any distribution $p$ on $\mathbb{R}^d$, $L^2(p)$ denotes the space of squared integrable random vectors in $\mathbb{R}^d$. We recall the definitions of the total variation distance $\mathrm{TV}(P, Q) = \frac{1}{2} \int_{\mathbb{R}^d} |p - q| \, dx$ for densities $p, q$, and the Wasserstein-2 distance $W_2^2(P, Q) = \inf_{\pi \in \Pi(P,Q)} \int \|x - y\|^2 \, d\pi(x, y)$ where $\Pi(P, Q)$ is the set of couplings. Finally, for any vector $x = (x_1, \ldots, x_d) \in \mathbb{R}^d$, $\|x\|_\infty = \max_i |x_i|$ and if $x$ is random, $\|x\|_{p,\infty} = \max_i (\mathbb{E}|x_i|^p)^{1/p}$.

### 2.1. Diffusion models

Here we review the basics of diffusion models; for a more detailed overview, we refer the reader to the surveys of Chen et al. (2024b) and Nakkiran et al. (2024).

Diffusion models are built upon two main components: the *forward process* and the *reverse process*. The forward process is a noise process driven by a stochastic differential equation of the form

$$dx_t = -x_t \, dt + \sqrt{2} \, dB_t, \qquad x_0 \sim q_{\text{pre}},$$

where $(B_t)_{t \geq 0}$ is a standard Brownian motion in $\mathbb{R}^d$. Conditioned on $x_0$, the process at time $t$ is distributed as $e^{-t} x_0 + \sqrt{1 - e^{-2t}} g$ for $g \sim \gamma^d$; denote this marginal distribution by $p_t^{\rightarrow}$, where $p_0^{\rightarrow} = q_{\text{pre}}$.

We run the forward process up to time $T$. Define

$$\sigma_t = \sqrt{1 - e^{-2(T-t)}}. \tag{1}$$

The *reverse process* is designed to undo this noise process, i.e. transform $p_T^{\rightarrow}$ to $p_0^{\rightarrow}$. For convenience of the notation, we denote $p_{T-t}^{\rightarrow}$ by $q_t$, so that $q_T = q_{\text{pre}}$. One version of this reverse process is given by the *probability flow ODE*, which is specified by

$$\mathrm{d}y_t = (y_t + \nabla \ln q_t(y_t)) \, \mathrm{d}t \, , \tag{2}$$

where $\nabla \ln q_t$ is called the *score function*. The key property is that if $y_0 \sim q_0$, then $y_T \sim q_T = q_{\text{pre}}$. In practice, this is run using *score estimates* $s_t \approx \nabla \ln q_t$ instead of the actual score functions, and in the theoretical literature it is standard to assume that these are close in $L_2(q_t)$. In this work, we will make a somewhat stronger assumption:

**Assumption 2** (Sub-exponential score error). *There is some parameter $\varepsilon_{\text{err}} > 0$ such that the error incurred by the* score *estimate $s_t : \mathbb{R}^d \to \mathbb{R}^d$ has sub-exponential norm at most $\varepsilon_{\text{err}}$ for all $0 \le t \le T$. That is, $\mathbb{P}_t[\|s_t - \nabla \ln q_t\| \ge z] \le 2 \exp(-z/\varepsilon_{\text{err}})$ for all $y \ge 0$.*

We leave as open whether this can be relaxed to the more standard assumption of $L_2$-accurate score estimation but note that even under the assumption of *perfect* score estimation, it was previously not known how to achieve high-accuracy guarantees.

Additionally, instead of initializing the ODE at $y_0 \sim q_0$, one initializes at $y_0 \sim \gamma^d$ for some noise distribution for which $\pi \approx q_0$ and from which it is easy to sample. For instance, in the example of the standard OU process, we can take $\pi = \gamma^d$ because the forward process converges exponentially quickly to $\gamma^d$. As $\pi$ is easy to sample from, the probability flow ODE can be used to (approximately) generate fresh samples from $q$.

Finally, we further assume the score estimate is Lipschitz, which is justified as we prove in the supplement that the true score is Lipschitz.

**Assumption 3** (Lipschitz score estimate). *$s_t$ is $\tilde{L}$-Lipschitz: $\forall x, y \in \mathbb{R}^d$, $\|s_t(x) - s_t(y)\| \le \tilde{L}\|x - y\|$.*

## 2.2. A fundamentally different view on discretization

Traditionally, to simulate the continuous-time ODE in discrete time, some numerical method like Euler-Maruyama discretization is used. Such discretization schemes can only approximately simulate the continuous time ODE by using a fixed gradient in a window of time. For example, to approximate the ODE $x_t = f_t(x_t)dt$ with one gradient step in time window $[0, h]$, the Euler-Maruyama method uses the gradient at the beginning time 0, resulting in the update $\hat{x}_h = hf_0(\hat{x}_0) + x_0$, where $\hat{x}_h$ denotes the discretized process. Here, $h$ is the step size of the algorithm. Unfortunately, this type of discretization puts a fundamental dimension-dependent limitation on how large the step size can be.

To illustrate the point, consider the vector field $f_t = x$ for all times $t \ge 0$. In this case, the discretized step is given by $\hat{x}_h = h(1)_{i=1}^d + x_0 = ((1+h))_{i=1}^d$ while the solution of the ODE $x_t = f_t(x_t)dt$ with initial condition $x_0 = (1)_{i=1}^d$ is given by $x_t = (e^t)_{i=1}^d$. This implies $\|x_h - \hat{x}_h\| = \|(e^h - (1+h))_{i=1}^d\| = \Theta(h^2\sqrt{d})$. Therefore, in order to keep the deviation $\|\hat{x}_h - x_h\|$ bounded by $O(1)$, the step size $h$ has to be $O(1/d^{1/4})$. This is because as we move along the ODE solution $x(t)$, the vector field $f_t(x_t)$ can change drastically from its initial value, so using a fixed vector $f_0(x_0)$ as the velocity does not allow us to go that far.

However, if we had a prior knowledge about the functional form of $f_t(x_t)$ along the path $(x_t)_{[0,h]}$, perhaps we could use that to take larger steps to simulate the ODE. One approach would be to leverage *higher-order smoothness of $f_t$* and run a higher-order numerical solver; as discussed in Section 1.2, many prior works pursued this approach (Li et al., 2023; Wu et al., 2024; Li et al., 2024a; Huang et al., 2025a;b; Li et al., 2025b), but none of them are known to achieve exponential acceleration in $\varepsilon$.

Instead, we opt for a different strategy. Suppose that we knew $f_t(x_t)$ lives in a low-dimensional subspace with basis $\{\phi_j(t)\}_{j=1}^D$, namely $f_t(x_t) = \sum_{i=1}^D c_i \phi_i$ for some coefficients $(c_i)_{i=1}^D \in \mathbb{R}^D$, then if we could estimate these coefficients, that allows us to predict the path $x_t$ for large times $t$. In this work, we show that perhaps surprisingly, under minimal conditions on the target distribution, each coordinate of the score function on the probability flow ODE path, $f_t(x_t)$, is well-approximated by a low-degree polynomial in $t$. We then prove that one can estimate these coefficients for each coordinate using the *collocation method* together with our score estimates. Notably, this enables us to take longer discretized steps, without any explicit dimension dependency, to follow the ODE path. We emphasize that while the collocation method, i.e. Picard iteration, has been used in prior diffusion model theory work (Gupta et al., 2024a; Chen et al., 2024a; Shih et al., 2024), the key novelty in our approach is to leverage low-degree polynomial approximation.

## 2.3. Collocation method

The *collocation method* is a numerical scheme for approximating the solution to an ordinary differential equation through fixed-point iteration. Here, consider a generic initial value problem: $\mathrm{d}x_t = f_t(x_t) \, \mathrm{d}t$ and $x_0 = v$ for all $t \in [0, H]$. This admits the integral representation $x_t = v + \int_0^t f_s(x_s) \, \mathrm{d}s$, which can be thought of as the fixed-point solution to the equation $x = \mathcal{T}(x)$ for the operator $\mathcal{T} : \mathcal{C}([0, H], \mathbb{R}^d) \to \mathcal{C}([0, H], \mathbb{R}^d)$ given by $\mathcal{T}(x)_t \triangleq v + \int_0^t f_s(x_s) \, \mathrm{d}s$ which has a unique fixed point provided there exists $k \in \mathbb{N}$ such that $\mathcal{T}^k$ is $L$-Lipschitz with $L < 1$. Lee et al. (2018) proposed to solve this fixed-

point equation as follows. Suppose that each coordinate of the time derivative of the solution, i.e. $t \mapsto f_t(x_t)$, is well-approximated by a low-degree polynomial. Let $c_1, \ldots, c_D \in [0, H]$, by polynomial interpolation we can find polynomials $\phi_1, \ldots, \phi_D : [0, H] \to \mathbb{R}$ such that $\phi_j(c_i) = \mathbb{1}[i = j]$ for all $i, j \in [D]$ so that in particular,

$$\frac{\mathrm{d}x_t}{\mathrm{d}t} \approx \sum_{j=1}^{D} f_{c_j}(x_{c_j})\phi_j(t) .$$

Writing this in integral form as before, we arrive at the approximate fixed-point equation:

$$x \approx \mathcal{T}_\phi(x) , \quad \mathcal{T}_\phi(x)_t \triangleq v + \int_0^t \sum_{j=1}^{D} f_{c_j}(x_{c_j})\phi_j(s) \, \mathrm{d}s . \quad (3)$$

This suggests a natural algorithm: instead of maintaining the entire continuous-time solution $x : [0, H] \to \mathbb{R}^d$ over the course of fixed-point iteration, simply maintain the values $(x_{c_j})_{j \in [D]}$ and update these according to Eq. (3), which amounts to a matrix-vector multiplication. This algorithm is summarized in Algorithm 1 below.

---

**Algorithm 1:** PICARD($(f_t)_{t \in [0, H]}, v, N$)

**Input:** Vector field $f_t : \mathbb{R}^d \to \mathbb{R}^d$ for $t \in [0, H]$, initial value $v$, number of iterations $N$
**Output:** Approximate solution $\hat{x}_H$ to initial value problem
1 Define $A_\phi \in \mathbb{R}^{D \times D}$ by $(A_\phi)_{i,j} = \int_0^{c_j} \phi_i(s) \, \mathrm{d}s$
2 Define $X^{(0)} \in \mathbb{R}^{d \times D}$ to be $v\mathbb{1}_D^\top$, where $\mathbb{1}_D$ is the all-ones vector.
3 **for** $t = 0, \ldots, N - 1$ **do**
4      Define $F_c(X^{(t)}) \in \mathbb{R}^{d \times D}$ by
        $F_c(X^{(t)})_{:,j} = f_{c_j}(X^{(t)}_{:,j})$
5      $X^{(t+1)} \leftarrow v\mathbb{1}_D^\top + F_c(X^{(t)})A_\phi$        // (3)
6 **end**
7 **return** $v + \int_0^H \sum_{i=1}^{D} F_{c_i}(X^{(N)}_{:,i})\phi_i(s) \, \mathrm{d}s$

---

Lee et al. (2018) give conditions under which collocation with appropriately chosen basis polynomials and nodes converges to a sufficiently accurate solution to the ODE. We will need to adapt their guarantees to our setting to account for score estimation error. The full guarantee will be given in Section 3. We require the following condition for the polynomial basis $\phi$:

**Definition 2.1.** *We say that $\phi$ is $\gamma_\phi$-bounded if, for all basis elements $\phi_j : [0, H] \to \mathbb{R}$ and $0 \le x \le H$, $\sum_{j=1}^{D} \left| \int_0^H \phi_j(s) \, \mathrm{d}s \right| \le \gamma_\phi t$.*

In this work, concretely, we will take $\phi_j(x) = \widetilde{\phi}_j(2x/H - 1)$, where $\widetilde{\phi}_j(x) = \frac{\sqrt{1 - c_j^2}\cos(D \cos^{-1} x)}{D(x - c_j)}$, where $c_j =$

$\cos(\frac{2j-1}{2D}\pi)$ is the $j$-th root of the Chebyshev polynomial of degree $D$. As shown in the proof of (Lee et al., 2018, Lemma 2.9), $\widetilde{\phi}_j$ is a degree-$j$ polynomial satisfying $\widetilde{\phi}_j(c_i) = \mathbb{1}[i = j]$; moreover, (Lee et al., 2018) notes that by (Lee & Vempala, 2017, Lemma 91), $\sum_{j=1}^{D} \left| \int_{-1}^{1} \widetilde{\phi}_j(x) \mathrm{d}x \right| \le 2000$. After a change of variable to rescale from $\widetilde{\phi}_j$ to $\phi_j$, we conclude that this basis $(\phi_j)$ is $\gamma_\phi = O(H)$-bounded.

### 2.4. Our algorithm

We are now ready to combine the ingredients in the previous subsections to give a description of our sampler. The algo-

---

**Algorithm 2:** COLLOCATIONDIFFUSION

**Input:** Score estimates $s_t$ satisfying Assumptions 2 and 3, target error $0 < \varepsilon < 1$, denoising schedule $0 < t_1 < \cdots < t_n = T$, Picard depth $N$
**Output:** Sample from a distribution $\hat{q}$ satisfying $W_2(\hat{q}, q) \le \varepsilon$
1 $\hat{x}_0 \sim \gamma^d$
2 **for** $k = 1, \ldots, n - 1, n$ **do**
3      $\hat{x}_{t_k} = $ PICARD($(s_t)_{t \in [t_{k-1}, t_k]}, \hat{x}_{t_{k-1}}, N$)
4 **end**
5 **return** $\hat{x}_{t_n}$

---

rithm simply splits up the reverse process into a series of small time windows, each of which is of length $\tilde{O}(\frac{\sigma^2}{R^2})$, and successfully solves the corresponding initial value problems in each of these windows using collocation via PICARD. There will be some accumulation of errors, as the PICARD calls do not result in exact solutions, but these errors can be made extremely small because PICARD solves the relevant ODEs to extremely high precision.

## 3. Overview of proof

Here we describe the main ingredients of our proof, deferring proofs to the supplement. Recall that key to our approach is to prove the structural result that the time derivative of the score along the reverse process can be controlled.

More precisely, Lemma 3.1 below *expresses* the first time derivative of the vector field along the trajectory. By iterating, we obtain Lemma 3.2 which *expresses* the $k^{\text{th}}$ time derivative of the vector field along the trajectory. By upper bounding the r.h.s. in Lemma 3.2, we obtain Lemma 3.3 which *bounds* the $k^{\text{th}}$ time derivative of the vector field along the trajectory. This bound means that the vector field along the trajectory has a small $k^{\text{th}}$ derivative, therefore is well approximated by a polynomial. To derive our method, we can therefore approximate the vector field along the trajectory

by a polynomial, and run collocation with this polynomial thanks to Proposition 3.6.

## 3.1. Low-degree approximation along the true reverse process

Our starting point is to relate time derivatives of the score at a point $y_t$ along the reverse process to higher-order moments of the posterior distribution on the clean sample that would have generated $y_t$ along the forward process.

We begin with the following calculation.

**Lemma 3.1.** *Suppose* $y_t = X_t + \sigma_t \xi$, *where* $X_t = e^{-(T-t)} \bar{X}$ *for* $\bar{X} \sim q_{\mathsf{pre}}$ *and* $\xi \sim \gamma^d$. *We use the notation* $\mathbb{E}^{t,y_t}$ *to denote the conditional expectation of* $X_t$ *given* $y_t = X_t + \sigma_t \xi$, *and* $q^{t,y_t}$ *to denote the density for this posterior. Define the posterior mean* $\mu_t(y) \triangleq \mathbb{E}^{t,y} X_t$.

*The time derivative of the vector field of probability flow ODE can be calculated as*

$$\partial_t(y_t + \nabla \log q_t(y_t)) = \left( y_t + \frac{\mathbb{E}^{t,y_t} X_t - y_t}{\sigma_t^2} \right) + \frac{1}{\sigma_t^2} \mathbb{E}^{t,y_t} X_t$$

$$+ \frac{1 - \sigma_t^2}{2\sigma_t^6} \mathbb{E}^{t,y_t} \left( \langle y_t, X_t \rangle + \|X_t\|^2 \right) (X_t - X_t')$$

$$+ \frac{4}{\sigma_t^4} \mathbb{E}^{t,y_t} \left( \langle y_t, X_t \rangle - \|X\|^2 \right) (X_t - X_t')$$

$$+ \frac{1}{\sigma_t^2} \left( \frac{1}{\sigma_t^2} - 1 \right) y_t - \frac{1}{\sigma_t^4} \mathbb{E}^{t,y_t} X_t$$

$$+ \frac{1}{\sigma_t^4} \mathbb{E}^{t,y_t} \left( 2 \left\langle y_t, \frac{X_t''}{\sigma_t^2} \right\rangle - \langle X_t, y_t \rangle \right.$$

$$\left. - \frac{1}{\sigma_t^2} \langle X_t, X_t'' \rangle \right) (X_t - X_t').$$

*Here,* $X_t, X_t', X_t''$ *denote independent, identically distributed draws from the posterior distribution with expectation operator* $\mathbb{E}^{t,y_t}$.

Iterating this calculation multiple times, we arrive upon the following key calculation expressing the higher-order derivatives of the vector field of the probability flow ODE:

**Lemma 3.2.** *The kth derivative of the backward ODE vector field can be expanded as*

$$\partial_t^k(y_t + \nabla \ln q_t(y_t))$$

$$= \sum_{r_1, r_2, r_3, r_4} \sum_{\mathbf{i} \in [k]^{r_3}, \mathbf{k} \in [k]^{r_4}, \mathbf{l} \in [k]^{r_4}} \frac{\left( 1 - \sigma_t^2 \right)^{r_1}}{\sigma_t^{r_2}}$$

$$\times \mathbb{E}^{t,y_t} \prod_{j=1}^{r_3} \left\langle y_t, X^{(\mathbf{i}_j)} \right\rangle \prod_{j=1}^{r_4} \left\langle X^{(\mathbf{k}_j)}, X^{(\mathbf{l}_j)} \right\rangle$$

$$\times \left( a_{\mathbf{i},\mathbf{k},\mathbf{l}} X + b_{\mathbf{i},\mathbf{k},\mathbf{l}} y_t \right),$$

*where* $r_1, r_2, r_3 \leq k$ *and* $r_2 \leq 4k$ *and the coefficients* $(a_{\mathbf{i},\mathbf{k},\mathbf{l}}, b_{\mathbf{i},\mathbf{k},\mathbf{l}})$ *satisfy* $\sum_{\mathbf{i} \in [k]^{r_3}, \mathbf{k} \in [k]^{r_4}, \mathbf{l} \in [k]^{r_4}} |a_{\mathbf{i},\mathbf{k},\mathbf{l}}| +$

$|b_{\mathbf{i},\mathbf{k},\mathbf{l}}| \leq (74k)^k$ . *Here the random variables* $X^{(i)}$ *are independently distributed according to the posterior distribution* $q^{t,y}$, *and* $\mathbf{i}, \mathbf{k}, \mathbf{l}$ *are arbitrary tuples of indices.*

This result can be understood in the same spirit as Tweedie's formula, which relates the posterior mean of $X$ to the score function. Higher-order generalizations of Tweedie's formula in the literature (Meng et al., 2021) typically relate moments of the posterior to derivatives of the score *with respect to the space variable*, whereas in Lemma 3.2 we consider derivatives with respect to the *time variable*.

An important feature of our technique is to bound this multiple integration formula for higher derivatives of the score independent of the ambient dimension $d$, which will be crucial for obtaining iteration complexity bounds that only depend on the effective radius $R$ of the distribution. More precisely, the bound on the effective radius allows us to bound expectations of the form $\mathbb{E}[\prod_{j=1}^{r_4} \langle X^{(\mathbf{k}_j)}, X^{(\mathbf{l}_j)} \rangle]$ pointwise and expectations of the form $\mathbb{E}[\prod_{j=1}^{r_3} \langle y, X^{(\mathbf{i}_j)} \rangle]$ with high probability by dimension-independent quantities, from which we can deduce bounds on the left-hand side of Lemma 3.2 via applications of Cauchy-Schwarz (see Lemma C.4). Leveraging these formulas, we arrive at our first key technical ingredient, a dimension-free bound on the higher-order time derivatives of the score:

**Lemma 3.3.** *If* $q$ *satisfies Assumption 1,* $(y_t)_t$ *is a solution to* (2), *and* $y_0 \sim q_T$, *then the* $k^{th}$ *derivative of the vector field along the probability flow ODE is bounded as*

$$\left\| \partial_t^k (y_t + \nabla \log q_t(y_t)) \right\|_{p,\infty}$$

$$\lesssim R \left( \frac{74k}{\sigma_t^2} \right)^k \left( \frac{R}{\sigma_t} + (kp)^{1/2} \right)^{2k}$$

Note that these bounds are singly exponential in the order of the derivative and scale with $(R/\sigma_t)^k$. If these derivative bounds held pointwise, then we could simply appeal to the Taylor remainder theorem to argue that the true score is well-approximated by a low-degree polynomial. But ultimately Lemma 3.3 only offers a high-probability statement over the distribution induced by the true reverse process. We will need to handle deviations from the true reverse process, which arise because the sampler is initialized at Gaussian instead of the correct marginal, and because in each subsequent window in which we apply collocation, the initialization is also slightly off from the correct marginal due to discretization errors accumulated from previous windows. Next, we describe how to deal with these deviations.

### 3.2. Low-degree approximation along the algorithm

To argue about low-degree approximation along the trajectory of the actual sampler, we will need to argue that the posterior distribution from Lemma 3.2 is robust to perturbations to the conditioning.

Our key tool for doing so is the following coupling lemma:

**Lemma 3.4** (Coupling). *Let $\tilde{y}$ a random variable such that $\|y - \tilde{y}\| \leq \delta$ for $\delta \leq \frac{1}{6(\sqrt{d} + \sqrt{\ln(1/\varepsilon_1)})}$. Then, given $T - t \geq 1$, under the event $\mathcal{A}_1$ defined in Equation (13), we have*

$$\mathrm{TV}(q^{t,y}, q^{t,\tilde{y}}) \leq 8\delta(\sqrt{d} + \sqrt{\ln(1/\varepsilon_1)}). \tag{4}$$

*In particular, Eq. (4) holds with probability at least $1 - \varepsilon_1$.*

Combining Lemmas 3.3 and 3.4, we obtain a bound on higher time derivatives of the score along the reverse process when initialized at a distribution that is slightly off from the true marginal:

**Theorem 3.5.** *Let $\delta \in (0,1)$. If $q$ satisfies Assumption 1, and $(\tilde{y}_t)_t$ is given by initializing the reverse process at a distribution which is $\delta$-close in $W_2$ to $q_0$ and running the probability flow ODE in continuous time. Then, for $\delta$ small enough (as rigorously characterized in Lemma E.6), with probability at least $1 - \delta_2$ over the randomness of the initialization, the $k^{th}$ derivative of the vector field along the probability flow ODE is bounded at $\tilde{y}_t$ by*

$$\left\| \partial_t^k \left( \tilde{y}_t + \nabla \log q_t(\tilde{y}_t) \right) \right\|_\infty$$
$$\lesssim R \left( \frac{k}{\sigma_t^2} \right)^k \left( \frac{R}{\sigma_t} \right)^{2k} + (k \log(74kd/\delta_2))^{2k}.$$

This theorem provides the theoretical underpinning for our approach. Because it is a high probability bound over iterates of a process initialized at the correct intermediate points of the true sampler, we can now instantiate the aforementioned Taylor truncation argument to get the desired low-degree approximation along the trajectory of the sampler itself, rather than of the idealized reverse process. We will use this ingredient in conjunction with the technology of Lee et al. (2018) to prove the following convergence bound for PICARD (Algorithm 1), recalling the notations from Section 2.3:

**Proposition 3.6** (Informal, see Appendix). *Suppose PICARD is run using a $\gamma_\phi$-bounded polynomial basis $\phi$ with nodes $(c_j)$. For any $h \leq \frac{1}{2\tilde{L}\gamma_\phi}$, for the true probability flow ODE $(y_t)$ and arbitrary $x \in \mathcal{C}([t_0, t_0 + h], \mathbb{R}^d)$,*

$$\left\| T_\phi^{\circ m}(y) - y \right\|_{[t_0, t_0 + h]} \leq 2 (1 + \gamma_\phi) h \cdot \Big( \varepsilon_{\mathsf{ld}} +$$
$$\max_{1 \leq j \leq D} \left\| s_{c_j}(y(c_j)) - y(c_j) - \nabla \log q_{c_j}(y(c_j)) \right\| \Big) \tag{5}$$

$$\left\| T_\phi^{\circ m}(x) - T_\phi^{\circ m}(y) \right\|_{[t_0, t_0 + h]} \leq \frac{1}{2^m} \left\| x - y \right\|_{[t_0, t_0 + h]} \tag{6}$$

*where here $\|\cdot\|_{[a,b]}$ denotes the sup norm over the interval $[a, b]$, and $\varepsilon_{\mathsf{ld}}$ denotes the approximation error (in sup norm) incurred by approximating the score function along the probability flow ODE with a polynomial in the span of $\phi$.*

Eq. (5) above shows that the trajectory of the true probability flow ODE, which is an exact fixed point for *exact* Picard iteration, is locally an approximate fixed point for the polynomial interpolation-based approximate Picard iteration given by PICARD (Algorithm 1). Eq. (6) shows that any trajectory which is locally close to this true trajectory contracts towards the true trajectory at an exponential rate, up until an error given by the bound in Eq. (5).

An important caveat is that the solver can only be run for time which scales inversely in the boundedness $\gamma_\phi$ of the basis (recall Definition 2.1), which can be of constant order, and inversely in the Lipschitzness of the vector field of the probability flow ODE. This is why we cannot directly use collocation to solve the probability flow ODE in one shot. Nevertheless, the crucial point is that, as the second inequality above suggests, the error incurred by PICARD is decreasing exponentially in the number of iterations, modulo some additional error that needs to be carried around coming from low-degree approximation and score estimation error. We will then chain together multiple applications of the above bound to get our final guarantee.

Finally, we note that while the core idea for the Proposition is from Lee et al. (2018), one novelty of our bound is that we account for score estimation error and show that the errors coming from that do not compound exponentially over the Picard iterations. This will be essential for keeping our sampler sufficiently close to the true trajectory of the probability flow ODE.

### 3.3. Convergence of the sampler

We now have all the necessary ingredients to prove our main result. We first show how to get a sampler which is $W_2$-close to $q$. An appealing feature of this guarantee relative to our TV convergence guarantee is that 1) the requirement on the score estimation error is far less stringent, and 2) it only involves running the algorithm COLLOCATIONDIFFUSION, i.e. no postprocessing is needed on top of simply simulating the true reverse process using the ODE solver of Lee et al. (2018) to implement discrete steps.

**Theorem 3.7.** *Let $\varepsilon_{\mathsf{err}} \leq O(\varepsilon / \ln(d/R))$, $\sigma > 0$ and $R \geq 1$. Suppose $q$ is a distribution satisfying Assumption 1 for parameters $R, \sigma$. Given access to score estimates $s_t$ satisfying Assumptions 2 and 3, with probability at least $1 - O(\varepsilon(R/\sigma)^2)$, Algorithm 2 outputs samples from a distribution $\hat{q}$ for which $W_2(q, \hat{q}) \leq \tilde{O}(\varepsilon \cdot \log^2(1/\varepsilon))$ in $\tilde{O}((R/\sigma)^2 \cdot \log(1/\varepsilon))$ number of rounds and $\tilde{O}(\log(1/\varepsilon))$ number of Picard iterations per round, for a total of $\tilde{O}((R/\sigma)^2 \cdot \log^2(1/\varepsilon))$ calls to the score estimate oracle.*

**Remark 3.8.** *Above, $\tilde{O}(\cdot)$ hides factors of $\log d, \log R, \log \tilde{L}, \log \log(1/\varepsilon)$. Furthermore, the assumption $R \geq 1$ is only for simplifying the bounds. For the full theorem please see Appendix G.*

Here we informally sketch the proof of this result, deferring the proof details to the Appendix. The idea will be to track the trajectory of the true probability flow ODE closely at all times and successively run PICARD over small time windows.

Suppose that up to some time $t$ in the simulation of the reverse process, the iterate of the sampler is distributed according to a distribution whose $W_2$ distance from the true marginal is small. Then consider trying to simulate the reverse process for another $h$ time steps to approximate $q_{T-t-h}$. One can do this by simply running COLLOCATIONDIFFUSION, provided $h$ is small enough, and because the starting distribution is $W_2$-close to the true marginal, the Taylor truncation argument in conjunction with the bound on the higher-order derivatives of the vector field ensure that we can safely invoke Proposition 3.6 and drive any error coming from the $W_2$ discrepancy exponentially quickly to zero, leaving behind a fixed amount of score estimation and polynomial approximation error.

Note that the exponential contraction of the error from the $W_2$ discrepancy is absolutely essential to our result. It allows us to "reset" the amount by which we stray from the curve of the true reverse process every time we run PICARD. This is reminiscent of a different recent approach to analyzing the probability flow ODE by Chen et al. (2024c). In that work, the authors tried simulating the ODE for short windows of time, appealing to a naive Wasserstein coupling argument in each of those windows. In their setting, the Wasserstein couplings incur some error that they would like to linearly, not exponentially, accumulate over the course of the sampler. The way to do this was to effectively "restart" the Wasserstein coupling at the start of each new time window by injecting a small amount of noise into the sampler. In contrast, rather than inject noise to restart the coupling, we use Picard iteration to suppress the accumulating Wasserstein error.

**Upgrading to total variation closeness.** We can use the above idea from Chen et al. (2024c) to derive an additional consequence of our main theorem. Specifically, as a consequence of regularizing properties of underdamped Langevin dynamics, our sampling guarantee in Wasserstein distance (Theorem 3.7) can be converted into a guarantee in total variation distance using standard arguments (Gupta et al., 2024a; Chen et al., 2024c), under the additional assumption that the true vanilla score $\nabla \ln q$ is Lipschitz.

**Corollary 3.9.** *Let $\varepsilon, R, \sigma > 0$. Suppose $q$ is a distribution satisfying Assumption 1 for parameters $R, \sigma$ and that $\nabla \ln q$ is $L$-Lipschitz. Given access to score estimates $s_t$ satisfying Assumption 2, with $\varepsilon_{\mathsf{err}} \leq \tilde{O}(\frac{\varepsilon^{2/3} L^{1/2} d^{1/6}}{(R/\sigma)^{2/3}})$, Algorithm 3 outputs samples from a distribution $\hat{q}$ for which $\mathrm{TV}(\hat{q}, q) \leq \varepsilon$ in $\tilde{O}\left((R/\sigma)^2\right)$ iterations.*

In the high-accuracy regime, it might seem counterintuitive how one can so easily convert to closeness in TV starting from closeness in Wasserstein, given that one has to run underdamped Langevin Monte Carlo without any kind of Metropolis adjustment. The key idea is to simply *run underdamped Langevin for a shorter amount of time* so that the bias coming from discretization is not too large, an idea from Gupta et al. (2024a). We make this argument formal in Appendix H.

## 4. Outlook

In this work, we gave a diffusion-based sampler for sampling from a wide class of distributions that includes Gaussian mixtures as a special case, which both has iteration complexity which scales polylogarithmically in $1/\varepsilon$ ("high-accuracy") and does not depend explicitly on the dimension but rather on the effective radius of the distribution. The key technical step was to establish a bound on the higher-order derivatives of the vector field of the probability flow ODE, which in turn allowed us to approximate it by a low-degree polynomial. This approximation result made it possible to exploit a variant of the collocation method, a fixed-point iteration for numerically solving ODEs, proposed by Lee et al. (2018) which converges exponentially quickly to the true ODE solution if run over sufficiently small time windows.

Our work has a few limitations that raise interesting questions for future study. Firstly, can we somehow relax our assumption on the underlying distribution $q$? Instead of insisting that it is a noised version of a distribution $q_{\mathsf{pre}}$ whose support is compact, we could merely ask that $q_{\mathsf{pre}}$ have bounded moments. We could also ask for TV or KL convergence to $q$ in a number of steps which scales only logarithmically in $1/\sigma$, matching what is known in the low-accuracy regime (Chen et al., 2023a; Benton et al., 2023). Lastly, is the assumption that the score errors have sub-exponential tails truly necessary for achieving high-accuracy?

## Impact statement

This paper presents work whose goal is to advance the field of machine learning. There are many potential societal consequences of our work, none of which we feel must be specifically highlighted here.

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

# A. Further related work

**Additional works on discretization bounds for diffusions.** There have been a large number works in recent years giving general convergence guarantees for diffusion models (De Bortoli et al., 2021; Block et al., 2022; De Bortoli, 2022; Lee et al., 2022; Liu et al., 2022; Pidstrigach, 2022; Wibisono & Yang, 2022; Chen et al., 2023b;c; Lee et al., 2023; Li et al., 2023; Benton et al., 2023; Chen et al., 2024c; Benton et al., 2024; Chen et al., 2023a; Gupta et al., 2024b; Li et al., 2024b; Li & Yan, 2024b; Conforti et al., 2025; Gupta et al., 2024a). Of these, one line of work (Chen et al., 2023b; Lee et al., 2023; Chen et al., 2023a; Benton et al., 2024; Conforti et al., 2025; Li & Yan, 2024b) analyzed DDPM, the stochastic analogue of the probability flow ODE, and showed $\tilde{O}(d)$ iteration complexity bounds, which were recently established by Benton et al. (2024); Conforti et al. (2025) under only the assumption that the data distribution has bounded second moment and very recently refined by Li & Yan (2024b) to apply to distributions with bounded first moment and to depend only linearly on $1/\varepsilon$. Another set of works (Chen et al., 2024c; 2023c; Li et al., 2023; 2024a; Gupta et al., 2024a; Li et al., 2024b; Li & Jiao, 2024; Jiao & Li, 2024) studied the probability flow ODE and proved similar rates, one notable difference being that under smoothness assumptions it is known how to get sublinear in $d$ dependence (Chen et al., 2024c; Gupta et al., 2024a; Li & Jiao, 2024; Jiao & Li, 2024). Recently, Zhang et al. (2025); Jiao et al. (2025) showed that sublinear scaling in $d$ is also possible purely using DDPMs. As it stands however, all of these works work only in the "low-accuracy" regime, that is, the best-known $\varepsilon$ achieved by these works is still $\text{poly}(1/\varepsilon)$, even under smoothness assumptions.

**High-accuracy sampling for log-concave distributions.** One of the central ideas in the line of work on high-accuracy log-concave sampling is to append a Metropolis-Hastings filter step after every step of an otherwise low-accuracy sampler. Combining this with Langevin Monte Carlo or Hamiltonian Monte Carlo gives rise to the popular methods of MALA or MHMC respectively. A line of works (Dwivedi et al., 2019; Chen et al., 2020; Lee et al., 2020; Chewi et al., 2021; Wu et al., 2022), culminating in the recent breakthrough of Altschuler & Chewi (2024), has established an iteration complexity of $\tilde{O}(\sqrt{d}\text{poly}(1/\varepsilon))$ for MALA for sampling log-concave distributions.

# B. Notation

We will work with a data distribution $q$ over $\mathbb{R}^d$. Throughout, $\bar{X}$ will denote a random vector sampled from $q$, and $X_t$ will denote the scaling $X_t = \sqrt{1 - \sigma_t^2}\bar{X}$. Likewise, we use $\bar{x}$ and $x_t$ to denote realizations of these random vectors. We use $\xi$ to denote a random vector sampled from $\gamma^d$. Given $t, y$, we will use $q^{t,y}$ to denote the posterior distribution on $X_t$ conditioned on $\sqrt{1 - \sigma_t^2}\bar{X} + \sigma_t\xi = y$. We will use $\mathbb{E}^{t,y}$ to denote expectation with respect to $q^{t,y}$, and write $\mu_t(y) \triangleq \mathbb{E}^{t,y}X_t$.

We will use the following noise schedule
$$\sigma_t \triangleq \sqrt{1 - e^{-2(T-t)}}, .$$
Throughout, we let $q_t \triangleq \text{law}(\sqrt{1 - \sigma_t^2}\bar{X} + \sigma_t\xi)$, so that $q_{\text{pre}} = q_T$.[1]

Given $t$, let $F_t^* : \mathbb{R}^d \to \mathbb{R}^d$ denote the map
$$F_t^*(y) = y + \nabla \ln q_t(y). \tag{7}$$
When $t$ is clear from context, we will omit the subscript. Note that
$$\nabla \ln q_t(y) = \frac{\mathbb{E}^{t,y}X_t - y}{\sigma_t^2} = \frac{\mu_t(y) - y}{\sigma_t^2}. \tag{8}$$

We denote our estimate for the score function by $F_t(y)$. When it is clear from the context, we drop the time index $t$ in $F_t$ and $F_t^*$. Moreover, we consider the following processes:

- $(y_t)$: the process given by the true probability flow ODE
$$dy_t = F_t^*(y_t)\, dt, \qquad y_0 \sim q_0$$

- $(\hat{y}_t)$: the process given by our Picard Algorithm 1 for $t \in [t_i, t_{i+1}]$, for all $1 \le i \le n$, as
$$\hat{y}_t \triangleq x_{t_i} + \int_0^{t-t_i} \sum_{i=1}^{D} F_{c_i}(X_{:,i})\phi_i(s)\, ds,$$

---

[1]Note that the convention in some prior works has been to parametrize time such that $q = q_0$, but we eschew this as our choice makes the notation in our proofs cleaner.

where $X_{:,i}$ are defined in Algorithm 1.

- $(\tilde{y}_t)$: the process that, in the interval $[t_i, t_{i+1}]$, is given by the true probability flow ODE starting from the algorithm iterate $\tilde{y}(t_i)$, for all $1 \leq i \leq n-1$, as defined in 2. Namely, $\tilde{y}(0) = y_{t_i}$, and for all $t \in [t_i, t_{i+1}]$:

$$\frac{d}{dt}\tilde{y}(t) = F_t^*(\tilde{y}(t)).$$

In the course of the proofs, we denote the interval $[t_i, t_{i+1}]$ for a fixed $1 \leq i \leq n-1$ by $[t_0, t_0 + h]$. For an arbitrary curve $z(t) : t \in [t_0, t_0 + h] \to \mathbb{R}^d$, we further use the following sup norm in our calculations:

$$\|z\|_{[t_0, t_0+h]} = \sup_{t_0 \leq t \leq t_0 + h} \|z(t)\|.$$

## C. Derivative calculations

In order to show that the score function $F_t^*(y_t)$ is a low-degree function of time $t$, our approach is to bound higher derivatives of $F_t^*(y_t)$ with respect to time. Having such bound then enables us to prove that $F_t^*(y_t)$ is close to its $k$th order tailor approximation using the Taylor remainder theorem, which we recall below.

**Fact C.1.** *For function $g : [t_0, t_0 + h] \to \mathbb{R}$, define the low degree approximation $P_{\leq k}g(s)$ with degree $k$ as:*

$$P_{\leq k}g(s) = \sum_{r=0}^{k-1} \frac{d^r g(s)}{ds^r}\Big|_{s=t_0} \frac{(s-t_0)^r}{r!}.$$

*Then by the Taylor remainder theorem, we have*

$$\|g - P_{\leq k}g\|_{[t_0, t_0+h]} \leq \frac{\max_{s \in [t_0, t_0+h]} \left|\frac{d^k g(s)}{ds^k}\right|}{k!} h^k.$$

With this in mind, we now proceed to prove dimension-independent bounds on the higher derivatives of the score function as stated in the main body (Lemmas 3.1-3.3).

The main building blocks of the bound in Lemma 3.2 are Lemmas C.1 and 3.1. Lemma C.1 has two parts, stated in Lemmas C.2, C.3 below which compute the derivative of the posterior mean $\mu_t(y)$ with respect to time ($t$) and space ($y$) variables. In Lemma 3.3 we combine all of these ingredients to obtain a bound on the higher derivatives of the score as a function of time.

### C.1. Proof of Lemma 3.1

We restate Lemma 3.1 here for convenience:

**Lemma 3.1.** *Suppose $y_t = X_t + \sigma_t \xi$, where $X_t = e^{-(T-t)}\bar{X}$ for $\bar{X} \sim q_{\text{pre}}$ and $\xi \sim \gamma^d$. We use the notation $\mathbb{E}^{t,y_t}$ to denote the conditional expectation of $X_t$ given $y_t = X_t + \sigma_t \xi$, and $q^{t,y_t}$ to denote the density for this posterior. Define the posterior mean $\mu_t(y) \triangleq \mathbb{E}^{t,y}X_t$.*

*The time derivative of the vector field of probability flow ODE can be calculated as*

$$\partial_t(y_t + \nabla \log q_t(y_t)) = \left(y_t + \frac{\mathbb{E}^{t,y_t}X_t - y_t}{\sigma_t^2}\right) + \frac{1}{\sigma_t^2}\mathbb{E}^{t,y_t}X_t$$

$$+ \frac{1-\sigma_t^2}{2\sigma_t^6}\mathbb{E}^{t,y_t}\left(\langle y_t, X_t\rangle + \|X_t\|^2\right)(X_t - X_t')$$

$$+ \frac{4}{\sigma_t^4}\mathbb{E}^{t,y_t}\left(\langle y_t, X_t\rangle - \|X\|^2\right)(X_t - X_t')$$

$$+ \frac{1}{\sigma_t^2}\left(\frac{1}{\sigma_t^2} - 1\right)y_t - \frac{1}{\sigma_t^4}\mathbb{E}^{t,y_t}X_t$$

$$+ \frac{1}{\sigma_t^4}\mathbb{E}^{t,y_t}\left(2\left\langle y_t, \frac{X_t''}{\sigma_t^2}\right\rangle - \langle X_t, y_t\rangle\right.$$

$$\left. - \frac{1}{\sigma_t^2}\langle X_t, X_t''\rangle\right)(X_t - X_t').$$

*Here, $X_t, X_t', X_t''$ denote independent, identically distributed draws from the posterior distribution with expectation operator $\mathbb{E}^{t,y_t}$.*

For this derivation, we will require the following calculation whose proof is given in Section C.2 below.

**Lemma C.1.** *In the notation of Lemma 3.1, for any $v \in \mathbb{R}^d$,*

$$\partial_t \mu_t(y) = \mu_t(y) + \frac{1-\sigma_t^2}{2\sigma_t^4} \mathbb{E}^{t,y} \left( \langle y, X_t \rangle + \|X_t\|^2 \right) (X_t - X_t') + \frac{4}{\sigma_t^2} \mathbb{E}^{t,y} \left( \langle y, X_t \rangle - \|X_t\|^2 \right) (X_t - X_t'),$$

$$D(\mu_t(y))[v] = \frac{1}{\sigma_t^2} \mathbb{E}^{t,y} \left( \langle y - X_t, v \rangle \right) (X_t - X_t'),$$

*where the expectations on the right-hand sides are with respect to i.i.d. replicas $X, X'$ sampled from the posterior distribution given $y$.*

The proof of Lemma 3.1 follows from Lemma C.1, namely combining Lemmas C.2 and C.3:

*Proof.* We can write

$$\partial_t(y_t + \nabla \ln q_t(y_t)) = y_t + \nabla \ln q_t(y_t) + \partial_t \nabla \ln q_t(y_t)$$
$$= y_t + \nabla \ln q_t(y_t) + \partial_t \nabla \ln q_t(y)\Big|_{y=y_t} + \nabla^2 q_t(y_t)(y_t + \nabla \ln q_t(y_t)).$$

Now using Lemma C.3 with $v = y_t + \nabla \ln q_t(y_t)$, as well as (8), we get

$$\nabla^2 q_t(y_t)(y_t + \nabla \ln q_t(y_t)) = \frac{1}{\sigma_t^2} \nabla \left( \frac{\mathbb{E}^{t,y} X_t - y}{\sigma_t^2} \right) \Big|_{y=y_t} [y_t + \nabla \ln q_t(y_t)]$$

$$= \frac{1}{\sigma_t^2} \left( -y_t + \frac{y_t - \mathbb{E}^{t,y_t} X_t}{\sigma_t^2} + \mathbb{E}^{t,y_t} \frac{1}{\sigma_t^2} \left( \langle y_t - X_t, y_t + \nabla \ln q_t(y_t) \rangle - \|y_t\|^2 + \frac{\|y_t\|^2}{\sigma_t^2} \right) (X_t - X_t') \right)$$

$$= \frac{1}{\sigma_t^2}(\frac{1}{\sigma_t^2} - 1)y_t - \frac{1}{\sigma_t^4} \mathbb{E}^{t,y_t} X_t + \frac{1}{\sigma_t^4} \mathbb{E}^{t,y_t} \left( 2 \left\langle y_t, \frac{X_t''}{\sigma_t^2} \right\rangle - \langle X_t, y_t \rangle - \frac{1}{\sigma_t^2} \langle X_t, X_t'' \rangle \right) (X_t - X_t').$$

Combining this with Lemma C.3, we obtain

$$\partial_t(y_t + \nabla \ln q_t(y_t)) = \partial_t y_t + \partial_t(\nabla \ln q_t(y))\Big|_{y=y_t} + \nabla^2 q_t(y_t)(y_t + \nabla \ln q_t(y_t))$$

$$= \partial_t y_t + \frac{1}{\sigma_t^2} \partial_t(\mathbb{E}^{t,y} X_t)\Big|_{y=y_t} + \nabla^2 q_t(y_t)(y_t + \nabla \ln q_t(y_t))$$

$$= \left( y_t + \frac{\mathbb{E}^{t,y_t} X_t - y_t}{\sigma_t^2} \right)$$

$$+ \frac{1}{\sigma_t^2} \mathbb{E}^{t,y} X_t + \frac{1-\sigma_t^2}{2\sigma_t^6} \mathbb{E}^{t,y} \left( \langle y, X_t \rangle + \|X_t\|^2 \right) (X_t - X_t') + \frac{4}{\sigma_t^4} \mathbb{E}^{t,y} \left( \langle y, X_t \rangle - \|X_t\|^2 \right) (X_t - X_t')$$

$$+ \frac{1}{\sigma_t^2}(\frac{1}{\sigma_t^2} - 1)y_t - \frac{1}{\sigma_t^4} \mathbb{E}^{t,y_t} X_t + \frac{1}{\sigma_t^4} \mathbb{E}^{t,y_t} \left( 2 \left\langle y_t, \frac{X_t''}{\sigma_t^2} \right\rangle - \langle X_t, y_t \rangle - \frac{1}{\sigma_t^2} \langle X_t, X_t'' \rangle \right) (X_t - X_t'). \quad (9)$$

where the last term is the result of taking derivative with respect to $\sigma_t$ in the denominator. $\qquad \square$

## C.2. Proof of Lemma C.1

Below we prove the two parts of Lemma C.1 computing the time and spatial derivatives of the posterior mean $\mu_t(y) = \mathbb{E}^{t,y} X_t$ respectively.

**Lemma C.2.** *Given any time $t$ and $y \in \mathbb{R}^d$, the time derivative of the expectation of the posterior distribution $q^{t,y}$ can be calculated as*

$$\partial_t \mathbb{E}^{t,y} X_t = \mathbb{E}^{t,y} X_t + \frac{1-\sigma_t^2}{2\sigma_t^4} \mathbb{E}^{t,y} \left( \langle y, X_t \rangle + \|X_t\|^2 \right) (X_t - X_t') + \frac{4}{\sigma_t^2} \mathbb{E}^{t,y} \left( \langle y, X_t \rangle - \|X_t\|^2 \right) (X_t - X_t'),$$

*where $X, X'$ are independent draws from the posterior distribution $q^{t,y}$.*

*Proof.* We have

$$\mathbb{E}^{t,y} X_t = \frac{\int e^{-\frac{\|y-x_t\|^2}{2\sigma_t^2}} x_t q_{\mathsf{pre}}(\bar{x}) d\bar{x}}{\int e^{-\frac{\|y-x_t\|^2}{2\sigma_t^2}} q_{\mathsf{pre}}(\bar{x}) d\bar{x}}$$

with random variables $\xi \sim \gamma^d$, $X_t = \sqrt{1-\sigma_t^2}\bar{X}_t$, and $\bar{X} \sim q_{\mathsf{pre}}$, where $y = X_t + \sigma_t \xi$ and we denote the values of $\bar{X}$ and $X_t$ by $\bar{x}$ and $x_t = \sqrt{1-\sigma_t^2}\bar{x}$, respectively. Then

$$
\partial_t \mathbb{E}^{t,y} X_t = \frac{\int e^{-\frac{\|y-x_t\|^2}{2\sigma_t^2}} x_t q_{\mathsf{pre}}(\bar{x}) d\bar{x}}{\int e^{-\frac{\|y-x_t\|^2}{2\sigma_t^2}} q_{\mathsf{pre}}(\bar{x}) d\bar{x}}
$$
$$
+ 4 \frac{\int e^{-\frac{\|y-x_t\|^2}{2\sigma_t^2}} x_t \frac{\langle y-x_t, x_t\rangle}{2\sigma_t^2} q_{\mathsf{pre}}(\bar{x}) d\bar{x}}{\int e^{-\frac{\|y-x_t\|^2}{\sigma_t^2}} q_{\mathsf{pre}}(\bar{x}) d\bar{x}} - 4 \frac{\int e^{-\frac{\|y-x_t\|^2}{2\sigma_t^2}} x_t q_{\mathsf{pre}}(\bar{x}) d\bar{x}}{\int e^{-\frac{\|y-x_t\|^2}{2\sigma_t^2}} q_{\mathsf{pre}}(\bar{x}) d\bar{x}} \frac{\int e^{-\frac{\|y-x_t\|^2}{2\sigma_t^2}} \frac{\langle y-x_t, x_t\rangle}{2\sigma_t^2} q_{\mathsf{pre}}(\bar{x}) d\bar{x}}{\int e^{-\frac{\|y-x_t\|^2}{\sigma_t^2}} q_{\mathsf{pre}}(\bar{x}) d\bar{x}}
$$
$$
+ \frac{\int e^{-\frac{\|y-x_t\|^2}{2\sigma_t^2}} x_t \frac{\|y-x_t\|^2}{2\sigma_t^4}(1-\sigma_t^2) q_{\mathsf{pre}}(\bar{x}) d\bar{x}}{\int e^{-\frac{\|y-x_t\|^2}{2\sigma_t^2}} q_{\mathsf{pre}}(\bar{x}) d\bar{x}} - \frac{\int e^{-\frac{\|y-x_t\|^2}{2\sigma_t^2}} x_t q_{\mathsf{pre}}(\bar{x}) d\bar{x}}{\int e^{-\frac{\|y-x_t\|^2}{2\sigma_t^2}} q_{\mathsf{pre}}(\bar{x}) d\bar{x}} \frac{\int e^{-\frac{\|y-x_t\|^2}{2\sigma_t^2}} \frac{\|y-x_t\|^2}{2\sigma_t^4}(1-\sigma_t^2) q_{\mathsf{pre}}(\bar{x}) d\bar{x}}{\int e^{-\frac{\|y-x_t\|^2}{2\sigma_t^2}} q_{\mathsf{pre}}(\bar{x}) d\bar{x}},
$$

where the first line follows from taking $\partial_t$ derivative w.r.t. $x_t$ that is not in the exponent (note that $x_t = \sqrt{1-\sigma_t^2}\bar{x} = e^{-(T-t)}\bar{x}$ is a function of $t$), the second line is the result of taking derivative w.r.t. $x_t$ that is in the exponent, and the final line is taking derivative w.r.t. the $1/\sigma_t^2$ in the exponent. Note that taking derivative w.r.t the numerator and then the denominator in these cases results in a covariance term between $x_t$ and $\langle y - x_t, x_t\rangle$ and $\|y - x_t\|^2$ in the second and third lines, respectively:

$$
\partial_t \mathbb{E}^{t,y} X_t = \mathbb{E}^{t,y} X_t + \mathrm{Cov}\left(\frac{(1-\sigma_t^2)\|y-X_t\|^2}{2\sigma_t^4}, X_t\right) + 4\,\mathrm{Cov}\left(\frac{\langle y-X_t, X_t\rangle}{\sigma_t^2}, X_t\right)
$$
$$
= \mathbb{E}^{t,y} X_t + \frac{1-\sigma_t^2}{2\sigma_t^4}\mathbb{E}^{t,y}\left(\langle y, X_t\rangle + \|X_t\|^2\right)\left(X_t - \mathbb{E}^{t,y} X_t\right) + \frac{4}{\sigma_t^2}\mathbb{E}^{t,y}\left(\langle y, X_t\rangle - \|X_t\|^2\right)\left(X_t - \mathbb{E}^{t,y} X_t\right)
$$
$$
= \mathbb{E}^{t,y} X_t + \frac{1-\sigma_t^2}{2\sigma_t^4}\mathbb{E}^{t,y}\left(\langle y, X_t\rangle + \|X_t\|^2\right)\left(X_t - X_t'\right) + \frac{4}{\sigma_t^2}\mathbb{E}^{t,y}\left(\langle y, X_t\rangle - \|X_t\|^2\right)\left(X_t - X_t'\right),
$$

where $X_t'$ is an independent replica of $X_t$. $\qquad\square$

**Lemma C.3.** *The directional derivative of the posterior mean with respect to $y$ is given by*

$$D(\mathbb{E}^{t,y} X_t)[v] = \frac{1}{\sigma_t^2}\mathbb{E}^{t,y}\left(\langle y-X_t, v\rangle\right)\left(X_t - X_t'\right) = \frac{1}{\sigma_t^2}\mathbb{E}^{t,y}\left(\langle y-X_t, v\rangle + \frac{\|y\|^2}{\sigma_t^2} - \|y\|^2\right)\left(X_t - X_t'\right).$$

*Proof.* We have

$$
D\left(\mathbb{E}^{t,y} X_t\right)[v] = -\frac{\int e^{-\frac{\|y-x_t\|^2}{2\sigma_t^2}}\left\langle\frac{y-x_t}{\sigma_t^2}, v\right\rangle x_t q_{\mathsf{pre}}(\bar{x}) d\bar{x}}{\int e^{-\frac{\|y-x_t\|^2}{2\sigma_t^2}} q_{\mathsf{pre}}(\bar{x}) d\bar{x}} + \frac{\int e^{-\frac{\|y-x_t\|^2}{2\sigma_t^2}} x_t q_{\mathsf{pre}}(\bar{x}) d\bar{x}}{\int e^{-\frac{\|y-x_t\|^2}{2\sigma_t^2}} q_{\mathsf{pre}}(\bar{x}) d\bar{x}} \frac{\int e^{-\frac{\|y-x_t\|^2}{2\sigma_t^2}}\left\langle\frac{y-x_t}{\sigma_t^2}, v\right\rangle q_{\mathsf{pre}}(\bar{x}) d\bar{x}}{\int e^{-\frac{\|y-x_t\|^2}{2\sigma_t^2}} q_{\mathsf{pre}}(\bar{x}) d\bar{x}}
$$
$$
= \mathrm{Cov}\left(\left\langle\frac{y-X_t}{\sigma_t^2}, v\right\rangle, X_t\right)
$$
$$
= \mathbb{E}^{t,y}\frac{1}{\sigma_t^2}\left(\langle y-X_t, v\rangle + \frac{\|y\|^2}{\sigma_t^2} - \|y\|^2\right)\left(X_t - \mathbb{E}^{t,y} X_t\right)
$$
$$
= \frac{1}{\sigma_t^2}\mathbb{E}^{t,y}\left(\langle y-X_t, v\rangle + \frac{\|y\|^2}{\sigma_t^2} - \|y\|^2\right)\left(X_t - X_t'\right),
$$

where we used the fact that $y$ is constant w.r.t the covariance calculation, hence adding $\frac{\|y\|^2}{\sigma_t^2} - \|y\|^2$ does not change the value of the covariance. $\qquad\square$

### C.3. Proof of Lemma 3.2

In this section, we prove the bound in Lemma 3.2 on the higher derivatives of the score function, restated here for convenience:

**Lemma 3.2.** *The $k$th derivative of the backward ODE vector field can be expanded as*

$$\partial_t^k \left(y_t + \nabla \ln q_t(y_t)\right)$$

$$= \sum_{r_1,r_2,r_3,r_4} \sum_{\mathbf{i}\in[k]^{r_3},\mathbf{k}\in[k]^{r_4},\mathbf{l}\in[k]^{r_4}} \frac{\left(1-\sigma_t^2\right)^{r_1}}{\sigma_t^{r_2}}$$

$$\times \mathbb{E}^{t,y_t} \prod_{j=1}^{r_3} \left\langle y_t, X^{(\mathbf{i}_j)} \right\rangle \prod_{j=1}^{r_4} \left\langle X^{(\mathbf{k}_j)}, X^{(\mathbf{l}_j)} \right\rangle$$

$$\times \left(a_{\mathbf{i},\mathbf{k},\mathbf{l}} X + b_{\mathbf{i},\mathbf{k},\mathbf{l}} y_t\right),$$

*where $r_1, r_2, r_3 \le k$ and $r_2 \le 4k$ and the coefficients $(a_{\mathbf{i},\mathbf{k},\mathbf{l}}, b_{\mathbf{i},\mathbf{k},\mathbf{l}})$ satisfy $\sum_{\mathbf{i}\in[k]^{r_3},\mathbf{k}\in[k]^{r_4},\mathbf{l}\in[k]^{r_4}} \left|a_{\mathbf{i},\mathbf{k},\mathbf{l}}\right| + \left|b_{\mathbf{i},\mathbf{k},\mathbf{l}}\right| \le (74k)^k$. Here the random variables $X^{(i)}$ are independently distributed according to the posterior distribution $q^{t,y}$, and $\mathbf{i}, \mathbf{k}, \mathbf{l}$ are arbitrary tuples of indices.*

*Proof.* To control the higher derivatives $\partial_t^k \left(y_t + \nabla \ln q_t(y_t)\right)$, Lemma 3.1 motivates us to analyze what happens after taking derivative from terms of the form

$$\frac{(1-\sigma_t^2)^{r_1}}{\sigma_t^{r_2}} \mathbb{E}^{t,y_t} \prod_{j=1}^{r_3} \left\langle y_t, X_t^{(\mathbf{i}_j)} \right\rangle \prod_{j=1}^{r_4} \left\langle X_t^{(\mathbf{k}_j)}, X_t^{(\mathbf{l}_j)} \right\rangle X_t, \tag{10}$$

$$\frac{(1-\sigma_t^2)^{r_1}}{\sigma_t^{r_2}} \mathbb{E}^{t,y_t} \prod_{j=1}^{r_3} \left\langle y_t, X_t^{(\mathbf{i}_j)} \right\rangle \prod_{j=1}^{r_4} \left\langle X_t^{(\mathbf{k}_j)}, X_t^{(\mathbf{l}_j)} \right\rangle y_t, \tag{11}$$

for some integers $r_1, r_2, r_3, r_4, r \le k$; such terms naturally appear after taking $k$ derivatives. In particular, we have

$$\partial_t \mathbb{E}^{t,y_t} X_t = \partial_t \mathbb{E}^{t,y} X_t \Big|_{y=y_t} + D(\mathbb{E}^{t,y} X_t)[y_t + \nabla \ln q_t(y_t)]$$

$$= \mathbb{E}^{t,y} X_t + \frac{1-\sigma_t^2}{2\sigma_t^4} \mathbb{E}^{t,y} \left(\langle y, X_t\rangle + \|X_t\|^2\right)(X_t - X_t') + \frac{4}{\sigma_t^2} \mathbb{E}^{t,y} \left(\langle y, X_t\rangle - \|X_t\|^2\right)(X_t - X_t')$$

$$+ \frac{1}{\sigma_t^2}\left(\frac{1}{\sigma_t^2} - 1\right) y_t - \frac{1}{\sigma_t^4} \mathbb{E}^{t,y_t} X_t + \frac{1}{\sigma_t^4} \mathbb{E}^{t,y_t} \left(2\left\langle y_t, \frac{X_t''}{\sigma_t^2}\right\rangle - \langle X_t, y_t\rangle - \frac{1}{\sigma_t^2} \langle X_t, X_t''\rangle\right)(X_t - X_t'),$$

and

$$\partial_t(y_t) = y_t + \nabla \ln q_t(y_t) = y_t + \frac{\mathbb{E}^{t,y_t} X_t - y_t}{\sigma_t^2}.$$

The key here is to keep track of the total number of $X_t$'s and $y_t$'s that appear in Equation (11). Let's denote this number by $\kappa = 2(r_3 + r_4)$. Then, first note that each step of taking derivatives, we are either taking derivative of some $y_t$ or some $X_t$ in Equation (11). In either case, the total number of $X_t$ and $y_t$'s increase by at most 2, and the power of $\sigma_t^2$ in the denominator, i.e. $r_2$, increases by at most 4. Hence, $\kappa \le 2k$. Taking derivative wrt $\sigma_t$:

$$\partial_t \frac{(1-\sigma_t^2)^{r_1}}{\sigma_t^{r_2}} = -r_2 \frac{(1-\sigma_t^2)^{r_1+1}}{\sigma_t^{r_2+2}} + \frac{-2r_1(1-\sigma_t^2)^{r_1-1}}{\sigma_t^{r_2-1}}, \tag{12}$$

which increases $r_1$ by at most one and $r_2$ by at most two. Therefore, $r_1 \le k$ and $r_2 \le 4k$. On the other hand, taking derivative with respect to either $y_t$ or $X_t$ creates at most $34 = 1 + 2 + 4 \times 2 \times 2 + 2 \times 2 + 2 + 1 + 4 \times 2$ new terms (counting term with coefficient 4 four times) and taking derivative wrt $\sigma_t$ (according to Equation (12)) creates at most $r_2 + 2r_1 \le 6k$, which means overall we have at most $(2 \times 34 + 6)k = 74k$ new terms generated after taking $k$ derivatives. Therefore, overall, after taking $k$ derivatives we have at most $(74k)^k$ terms generated. $\qquad\square$

## C.4. Proof of Lemma 3.3

We formally state and prove our upper bound on the derivatives $\partial_t^k (y_t + \nabla \ln q_t(y_t))$ in Lemma 3.3, restated here for convenience:

**Lemma 3.3.** *If $q$ satisfies Assumption 1, $(y_t)_t$ is a solution to (2), and $y_0 \sim q_T$, then the $k^{th}$ derivative of the vector field along the probability flow ODE is bounded as*

$$\left\| \partial_t^k \left( y_t + \nabla \log q_t(y_t) \right) \right\|_{p,\infty}$$
$$\lesssim R \left( \frac{74k}{\sigma_t^2} \right)^k \left( \frac{R}{\sigma_t} + (kp)^{1/2} \right)^{2k}$$

*Proof.* The proof directly follows from Lemma 3.2 in conjunction with Lemma C.4 below. $\square$

In Lemma C.4 we bound the $p$-norm of the terms that appear in Lemma 3.2.

**Lemma C.4.** *We have*

$$\left\| \frac{(1 - \sigma_t^2)^{r_1}}{\sigma_t^{r_2}} \prod_{j=1}^{r_3} \left\langle y_t, X_t^{(\mathbf{i}_j)} \right\rangle \prod_{j=1}^{r_4} \left\langle X_t^{(\mathbf{k}_j)}, X_t^{(\mathbf{l}_j)} \right\rangle X_t \right\|_{p,\infty} \vee \left\| \frac{(1 - \sigma_t^2)^{r_1}}{\sigma_t^{r_2}} \prod_{j=1}^{r_3} \left\langle y, X_t^{(\mathbf{i}_j)} \right\rangle \prod_{j=1}^{r_4} \left\langle X_t^{(\mathbf{k}_j)}, X_t^{(\mathbf{l}_j)} \right\rangle y_t \right\|_{p,\infty}$$

$$\leq R \left( \frac{1}{\sigma_t^2} \right)^k \left( \frac{R}{\sigma_t} + (kp)^{1/2} \right)^{2k},$$

*where by $\|v\|_{p,\infty}$ we mean the infinity norm of the $p$th moment of vector $v$.*

*Proof.* First, from the fact that the radius of support of $q_0$ is bounded by $R$, we have

$$\left\| \frac{(1 - \sigma_t^2)^{r_1}}{\sigma_t^{r_2}} \prod_{j=1}^{r_3} \left\langle y_t, X_t^{(\mathbf{i}_j)} \right\rangle \prod_{j=1}^{r_4} \left\langle X_t^{(\mathbf{k}_j)}, X_t^{(\mathbf{l}_j)} \right\rangle X_t \right\|_{p,\infty} \leq \frac{1}{\sigma_t^{r_2}} \left\| \prod_{j=1}^{r_3} \left\langle y_t, X_t^{(\mathbf{i}_j)} \right\rangle \right\|_p R^{2r_4+1}.$$

Now using the derivation in Lemma 4.4 in Gatmiry et al. (2024), we get

$$\left\| \prod_{j=1}^{r_3} \left\langle y_t, X_t^{(\mathbf{i}_j)} \right\rangle \right\|_p \leq R^{2r_3} \left( 1 + \frac{(r_3 p)^{1/2} \sigma_t}{R} \right)^{r_3}.$$

Plugging this above and using $R \geq 1$:

$$\text{LHS} \leq \frac{R^{2r_3 + 2r_4 + 1}}{\sigma_t^{r_2}} \left( 1 + \frac{(r_3 p)^{1/2} \sigma_t}{R} \right)^{r_3} \leq R \left( \frac{R}{\sigma_t^2} \right)^{2k} \left( 1 + \frac{(kp)^{1/2} \sigma_t}{R} \right)^k$$

$$\leq R \left( \frac{1}{\sigma_t} \right)^{2k} \left( \frac{R}{\sigma_t} \right)^{2k} \left( 1 + \frac{(kp)^{1/2} \sigma_t}{R} \right)^{2k}$$

$$\leq R \frac{1}{\sigma_t^{2k}} \left( \frac{R}{\sigma_t} + (kp)^{1/2} \right)^{2k}$$

Similarly for the other term, using Cauchy-Schwarz

$$\left\| \frac{(1 - \sigma_t^2)^{r_1}}{\sigma_t^{r_2}} \prod_{j=1}^{r_3} \left\langle y, X_t^{(\mathbf{i}_j)} \right\rangle \prod_{j=1}^{r_4} \left\langle X_t^{(\mathbf{k}_j)}, X_t^{(\mathbf{l}_j)} \right\rangle y \right\|_{p,\infty} \leq \frac{1}{\sigma_t^{r_2}} \left\| \prod_{j=1}^{r_3} \left\langle y, X_t^{(\mathbf{i}_j)} \right\rangle \right\|_{2p} R^{2r_4} \|y\|_{2,\infty}$$

$$\leq \frac{1}{\sigma_t^{r_2 - 1}} R^{2r_3} \left( 1 + \frac{(2r_3 p)^{1/2} \sigma_t}{R} \right)^{r_3} R^{2r_4}$$

$$\leq \frac{1}{\sigma_t^{2k}} \left( \frac{R}{\sigma_t} + (kp)^{1/2} \right)^{2k}. \qquad \square$$

## D. Proof of Lemma 3.4

We prove Theorem 3.5 by translating the inequality in Lemma 3.3 from $y$ to a close by $\tilde{y}$ via a coupling between the distribution of $X_t$ conditioned on $y$ and $\tilde{y}$. Note that the argument of Lemma 3.3 is not immediate for $\tilde{y}$ because it does not follow the true reverse process as $y$ does.

First, recall that $\bar{X} \sim q_{\mathsf{pre}}$ and $\xi \sim \gamma^d$ and $y_t = X_t + \sigma_t \xi$, $X_t = \sqrt{1 - \sigma_t^2} \bar{X}$. Now for every $\varepsilon_1 > 0$, consider the event

$$\mathcal{A}_1 := \left\{ \|y_t - X_t\| \leq \sqrt{d} + \sqrt{\ln(1/\varepsilon_1)} \right\}, \tag{13}$$

and note that $\mathbb{P}(\mathcal{A}_1) \geq 1 - \varepsilon_1$, since $\sigma_t \leq 1$.

**Lemma 3.4** (Coupling). *Let $\tilde{y}$ a random variable such that $\|y - \tilde{y}\| \leq \delta$ for $\delta \leq \frac{1}{6(\sqrt{d} + \sqrt{\ln(1/\varepsilon_1)})}$. Then, given $T - t \geq 1$, under the event $\mathcal{A}_1$ defined in Equation (13), we have*

$$\mathrm{TV}(q^{t,y}, q^{t,\tilde{y}}) \leq 8\delta(\sqrt{d} + \sqrt{\ln(1/\varepsilon_1)}). \tag{4}$$

*In particular, Eq. (4) holds with probability at least $1 - \varepsilon_1$.*

*Proof.* Since $T - t \geq 1$, we have

$$\sigma_t^2 = 1 - e^{-2(T-t)} \geq 1 - e^{-2} \geq 0.5.$$

Let $p(x, y)$ denote the joint distribution of $(X_t, y_t)$, for $y_t = X_t + \xi$, $X_t = \sqrt{1 - \sigma_t^2} \bar{X}$, $\bar{X} \sim q_{\mathsf{pre}}$, and $\xi \sim N(0, \sigma_t^2 I)$. Note that $q^{t,y} = p(\cdot \mid y)$. We have

$$\begin{aligned}
\left| \ln\left( \frac{p(y \mid x)}{p(\tilde{y} \mid x)} \right) \right| &= \frac{1}{2\sigma_t^2} \left| \|y - x\|^2 - \|\tilde{y} - x\|^2 \right| \\
&\leq \left| \|y - x\|^2 - \|\tilde{y} - x\|^2 \right| \\
&\leq 2\|y - \tilde{y}\| \|y - x\| + \|y - \tilde{y}\|^2 \\
&\leq 2\delta(\sqrt{d} + \sqrt{\ln(1/\varepsilon_1)}) + \delta^2 \\
&\leq 3\delta(\sqrt{d} + \sqrt{\ln(1/\varepsilon_1)}),
\end{aligned}$$

and by the assumed bound on $\delta$, we have $2\delta(\sqrt{d} + \sqrt{\ln(1/\varepsilon_1)}) + \delta^2 \leq 0.5$, which implies

$$(1 - 3\delta(\sqrt{d} + \sqrt{\ln(1/\varepsilon_1)}))p(y \mid x) \leq p(\tilde{y} \mid x) \leq p(y \mid x)(1 + 3\delta(\sqrt{d} + \sqrt{\ln(1/\varepsilon_1)})).$$

Multiplying $p(x)$ on both sides, we get

$$(1 - 3\delta(\sqrt{d} + \sqrt{\ln(1/\varepsilon_1)}))p(y \mid x)p(x) \leq p(\tilde{y} \mid x)p(x) \leq p(y \mid x)p(x)(1 + 3\delta(\sqrt{d} + \sqrt{\ln(1/\varepsilon_1)})).$$

But from Bayes' rule, it is easy to see that this implies

$$(1 - 3\delta(\sqrt{d} + \sqrt{\ln(1/\varepsilon_1)}))^2 q^{t,y}(x) \leq q^{t,\tilde{y}}(x) \leq q^{t,y}(x)(1 + 3\delta(\sqrt{d} + \sqrt{\ln(1/\varepsilon_1)}))^2.$$

Given that $(1 + 3\delta(\sqrt{d} + \sqrt{\ln(1/\varepsilon_1)}))^2 \leq 1 + 8\delta(\sqrt{d} + \sqrt{\ln(1/\varepsilon_1)})$ by the assumed bound on $\delta$, we conclude that

$$\mathrm{TV}(q^{t,y}, q^{t,\tilde{y}}) \leq 8\delta(\sqrt{d} + \sqrt{\ln(1/\varepsilon_1)}). \qquad \square$$

## E. Proof of Theorem 3.5

Here we show a similar bound on the derivatives of the vector field for the curve $\tilde{y}$ instead of $y$, where $y$ and $\tilde{y}$ are defined in Section B. Throughout this section, we assume $\tilde{y}_t$ and $y_t$ are close by $\delta$ in the window $[t_0, t_0 + h]$:

$$\|y_t - \tilde{y}_t\|_{[t_0, t_0+h]} \leq \delta. \tag{14}$$

Later on in the proof of Theorem G.1, we will show by induction that it is valid to assume Equation (14) holds. When it is clear, we will drop the $t$ indices from the variables for clarity. First, we restate Theorem 3.5 and show the short proof based on the rest of this section. We then get into the details of proving the intermediate lemmas.

**Theorem 3.5.** *Let $\delta \in (0,1)$. If $q$ satisfies Assumption 1, and $(\tilde{y}_t)_t$ is given by initializing the reverse process at a distribution which is $\delta$-close in $W_2$ to $q_0$ and running the probability flow ODE in continuous time. Then, for $\delta$ small enough (as rigorously characterized in Lemma E.6), with probability at least $1 - \delta_2$ over the randomness of the initialization, the $k^{th}$ derivative of the vector field along the probability flow ODE is bounded at $\tilde{y}_t$ by*

$$\left\| \partial_t^k \left( \tilde{y}_t + \nabla \log q_t(\tilde{y}_t) \right) \right\|_\infty$$
$$\lesssim R \left( \frac{k}{\sigma_t^2} \right)^k \left( \frac{R}{\sigma_t} \right)^{2k} + \left( k \log(74kd/\delta_2) \right)^{2k} .$$

*Proof.* For simplicity of exposition we will drop the time index $t$ from $y_t$, $X_t$ in the rest of this section, and write $y = X + \sigma_t \xi$, where $\bar{X} \sim q_{\mathrm{pre}}$, $X = \sqrt{1 - \sigma_t^2} \bar{X}$ and $\xi \sim \gamma^d$. Let $p(.|y)$ be the distribution of $X$ conditioned on $y$ and let $\tilde{X}$ be a sample from the conditional $p(.|\tilde{y})$. For index $i$ let $X^{(i)}$ and $\tilde{X}^{(i)}$ be an independent and identical sample of $X$ and $\tilde{X}$, respectively. Now using Lemma E.6 with $\delta_1 = \frac{\delta_2}{d(60k)^k}$, we get

$$\mathbb{P} \left( \left| \mathbb{E}_{\{\tilde{X}^{(i)}\}} \frac{(1 - \sigma_t^2)^{r_1}}{\sigma_t^{r_2}} \prod_{j=1}^{r_3} \left\langle \tilde{y}, \tilde{X}^{(\mathbf{i}_j)} \right\rangle \prod_{j=1}^{r_4} \left\langle \tilde{X}^{(\mathbf{k}_j)}, \tilde{X}^{(\mathbf{l}_j)} \right\rangle X \right|_\infty \geq eR \left( \frac{1}{\sigma_t^2} \right)^k \left( \frac{R}{\sigma_t} \right)^{2k} + e \left( k \log(74kd/\delta_2) \right)^k \right) \leq \delta_2 .$$

Taking union bound over all the terms, we get with probability at least $1 - \delta_2$

$$\left\| \partial_t^k \left( \tilde{y}_t + \nabla \log q_t(\tilde{y}_t) \right) \right\|_\infty \leq e \left( R + \delta + \sigma_t + 1 \right) \left( \frac{kh}{\sigma_t^2} \right)^k \left( \frac{R}{\sigma_t} \right)^{2k} + e \left( k \log(74kd/\delta_2) \right)^{2k} . \qquad \square$$

Next, we derive a pessimistic bound on $\mathbb{E} \left| \prod_{j=1}^{r_3} \left\langle \tilde{y}, X^{(\mathbf{i}_j)} \right\rangle \right|$ for all ordered tuples of indices $\mathbf{i}$, where each $X^{(i)}$ denotes an independent sample from the posterior distribution $q^{t,y}$, by obtaining a pessimistic bound on the term $\left\| \prod_{j=1}^{r_3} \left\langle y, X^{(\mathbf{i}_j)} \right\rangle \right\|_p$.

**Lemma E.1.** *For all tuples of indices $\mathbf{i}$, we have*

$$\mathbb{E} \left| \prod_{j=1}^{r_3} \left\langle \tilde{y}, X^{(\mathbf{i}_j)} \right\rangle \right| \leq \left( R^2 \left( 1 + \frac{(r_3 p)^{1/2} \sigma_t}{R} \right) + \delta R \right)^{r_3}$$

*Proof.* Using Cauchy-Schwarz,

$$\left\| \prod_{j=1}^{r_3} \left\langle \tilde{y}, X^{(\mathbf{i}_j)} \right\rangle \right\|_p \leq \sum_{S \subseteq [k]} \left\| \prod_{j \in S} \left\langle y, X^{(\mathbf{i}_j)} \right\rangle \right\|_p \left\| \prod_{j \notin S} \left\langle \tilde{y} - y, X^{(\mathbf{i}_j)} \right\rangle \right\|_p$$
$$\leq \sum_{\ell=1}^{r_3} \binom{r_3}{\ell} \left( R^2 \left( 1 + \frac{(r_3 p)^{1/2} \sigma_t}{R} \right) \right)^\ell (\delta R)^{r_3 - \ell}$$
$$\leq \left( R^2 \left( 1 + \frac{(r_3 p)^{1/2} \sigma_t}{R} \right) + \delta R \right)^{r_3} . \qquad \square$$

**Lemma E.2.** *For $\nu \triangleq \left( 3\delta(\sqrt{d} + \sqrt{\ln(1/\varepsilon_1)}) \right)^{1/p}$, we have for all tuples of indices $\mathbf{i}$ that*

$$\left\| \mathbb{1}\{\mathcal{A}_1\} \prod_{j=1}^{r_3} \left\langle \tilde{y}, \tilde{X}^{(\mathbf{i}_j)} \right\rangle \right\|_p \leq \left( R^2 \left( 1 + \frac{(2r_3 p)^{1/2} \sigma_t}{R} \right) + \delta R + \nu R \left( \sqrt{2pd} + R + \delta \right) \right)^{r_3} . \qquad (15)$$

*Proof.* First, note that we have the following bound:

$$
\left\| \prod_{j=1}^{r_3} \left\langle \tilde{y}, \tilde{X}^{(\mathbf{i}_j)} \right\rangle \right\|_p^p = \mathbb{E} \prod_{j=1}^{r_3} \left\langle \tilde{y}, \tilde{X}^{(\mathbf{i}_j)} \right\rangle^p
$$

$$
\leq \mathbb{E} \prod_{j=1}^{r_3} \|\tilde{y}\|^p \left\| \tilde{X}^{(\mathbf{i}_j)} \right\|^p
$$

$$
\leq \mathbb{E} \prod_{j=1}^{r_3} \left( \|y - \tilde{y}\| + \|\xi\| + \|X^{\mathbf{i}_j}\| \right)^p \left\| \tilde{X}^{(\mathbf{i}_j)} \right\|^p
$$

$$
\leq \mathbb{E} \prod_{j=1}^{r_3} \left( \delta + \|\xi\| + R \right)^p R^p
$$

$$
\leq \mathbb{E} \sum_{\ell=1}^{r_3 p} \binom{r_3 p}{\ell} \|\xi\|^\ell (R + \delta)^{r_3 p - \ell} R^{r_3 p}
$$

$$
\leq \sum_{\ell=1}^{r_3 p} \binom{r_3 p}{\ell} \left( \sqrt{\ell d} \right)^\ell (R + \delta)^{r_3 p - \ell} R^{r_3 p}
$$

$$
= \left( R \left( \sqrt{pd} + R + \delta \right) \right)^{r_3 p}, \tag{16}
$$

which implies

$$
\left\| \prod_{j=1}^{r_3} \left\langle \tilde{y}, \tilde{X}^{(\mathbf{i}_j)} \right\rangle \right\|_p \leq \left( R \left( \sqrt{pd} + R + \delta \right) \right)^{r_3}.
$$

Therefore, applying Lemma 3.4 with $\varepsilon_1 = \delta$, on event $\mathcal{A}_1$ we have

$$
TV(q^{t,y}, q^{t,\tilde{y}}) \leq 3\delta(\sqrt{d} + \sqrt{\ln(1/\varepsilon_1)}),
$$

which means for all $m$,

$$
TV(X^{(m)}, \tilde{X}^{(m)}) \leq 3\delta(\sqrt{d} + \sqrt{\ln(1/\varepsilon_1)}).
$$

Therefore, the optimal coupling between $X^{(m)}$ and $\tilde{X}^{(m)}$ conditioned on $y$ satisfies

$$
\mathbb{P}\left( X^{(m)} = \tilde{X}^{(m)} \right) \geq 1 - 3\delta(\sqrt{d} + \sqrt{\ln(1/\varepsilon_1)}) \tag{17}
$$

Now equipped with this coupling, for a subset $\mathcal{U} \subset [r_3]$, let $\mathcal{A}_2 = \mathcal{A}_2(\mathcal{U})$ be the event that $X^{(\mathbf{i}_j)} = \tilde{X}^{(\mathbf{i}_j)}$ for all $j \in \mathcal{U}$. Then we can write

$$
\mathbb{E}\left( \mathbb{1}\left\{ \mathcal{A}_1 \cap \mathcal{A}_2(\mathcal{U}) \right\} \prod_{j=1}^{r_3} \left\langle \tilde{y}, \tilde{X}^{(\mathbf{i}_j)} \right\rangle \right)^p
$$

$$
= \mathbb{E}\left( \mathbb{1}\left\{ \mathcal{A}_1 \cap \mathcal{A}_2(\mathcal{U}) \right\} \prod_{j \in \mathcal{U}} \left\langle \tilde{y}, \tilde{X}^{(\mathbf{i}_j)} \right\rangle \right)^p \left( \mathbb{1}\left\{ \mathcal{A}_1 \cap \mathcal{A}_2(\mathcal{U}) \right\} \prod_{j \notin \mathcal{U}} \left\langle \tilde{y}, \tilde{X}^{(\mathbf{i}_j)} \right\rangle \right)^p.
$$

$$
\leq \sqrt{ \mathbb{E}\left( \mathbb{1}\left\{ \mathcal{A}_1 \cap \mathcal{A}_2(\mathcal{U}) \right\} \prod_{j \in \mathcal{U}} \left\langle \tilde{y}, \tilde{X}^{(\mathbf{i}_j)} \right\rangle \right)^{2p} } \sqrt{ \mathbb{E}\left( \mathbb{1}\left\{ \mathcal{A}_1 \cap \mathcal{A}_2(\mathcal{U}) \right\} \prod_{j \notin \mathcal{U}} \left\langle \tilde{y}, \tilde{X}^{(\mathbf{i}_j)} \right\rangle \right)^{2p} } \tag{18}
$$

But on one hand, from Equation (17) and the independence of the couplings, we have

$$
\mathbb{P}(\mathcal{A}_2(\mathcal{U}) | \mathcal{A}_1) \leq \left( 3\delta(\sqrt{d} + \sqrt{\ln(1/\varepsilon_1)}) \right)^{r_3 - |\mathcal{U}|}.
$$

Therefore, similar to the derivation in (16)

$$
\mathbb{E}\left(1\{\mathcal{A}_1 \cap \mathcal{A}_2(\mathcal{U})\} \prod_{j \notin \mathcal{U}} \left\langle \tilde{y}, \tilde{X}^{(\mathbf{i}_j)} \right\rangle \right)^p
$$

$$
\leq \mathbb{E}1\{\mathcal{A}_1\} \sum_{\ell=1}^{(r_3-|\mathcal{U}|)p} \binom{(r_3-|\mathcal{U}|)p}{\ell} \|\xi\|^\ell (R+\delta)^{(r_3-|\mathcal{U}|)p-\ell} R^{(r_3-|\mathcal{U}|)p} \mathbb{P}_{|y}(\mathcal{A}_2(\mathcal{U})|\mathcal{A}_1)
$$

$$
\leq \mathbb{P}_{|y}(\mathcal{A}_2(\mathcal{U})|\mathcal{A}_1) \sum_{\ell=1}^{(r_3-|\mathcal{U}|)p} \binom{(r_3-|\mathcal{U}|)p}{\ell} \|\xi\|^\ell (R+\delta)^{(r_3-|\mathcal{U}|)p-\ell} R^{(r_3-|\mathcal{U}|)p}
$$

$$
\leq \left(3\delta(\sqrt{d}+\sqrt{\ln(1/\varepsilon_1)})\right)^{|\mathcal{U}|} \left(R\left(\sqrt{pd}+R+\delta\right)\right)^{(r_3-|\mathcal{U}|)p}
$$

$$
= \nu^{p(r_3-|\mathcal{U}|)} \left(R\left(\sqrt{pd}+R+\delta\right)\right)^{(r_3-|\mathcal{U}|)p},
$$

where $\nu \triangleq \left(3\delta(\sqrt{d}+\sqrt{\ln(1/\varepsilon_1)})\right)^{1/p}$. This implies

$$
\left\| 1\{\mathcal{A}_1 \cap \mathcal{A}_2(\mathcal{U})\} \prod_{j \notin \mathcal{U}}^{r_3} \left\langle \tilde{y}, \tilde{X}^{(\mathbf{i}_j)} \right\rangle \right\|_p \leq \left(3\delta(\sqrt{d}+\sqrt{\ln(1/\varepsilon_1)})\right)^{1/p} \left(R\left(\sqrt{pd}+R+\delta\right)\right)^{r_3-|\mathcal{U}|}. \tag{19}
$$

For the first term, we can use the coupling and substitute $\tilde{X}^{(\mathbf{i}_j)}$'s by $X^{(\mathbf{i}_j)}$ and then use Lemma E.1:

$$
\mathbb{E}\left(1\{\mathcal{A}_1 \cap \mathcal{A}_2(\mathcal{U})\} \prod_{j \in \mathcal{U}} \left\langle \tilde{y}, \tilde{X}^{(\mathbf{i}_j)} \right\rangle \right)^p = \mathbb{E}\left(1\{\mathcal{A}_1 \cap \mathcal{A}_2(\mathcal{U})\} \prod_{j \in \mathcal{U}} \left\langle \tilde{y}, X^{(\mathbf{i}_j)} \right\rangle \right)^p
$$

$$
\leq \mathbb{E}\left(\prod_{j \in \mathcal{U}} \left\langle \tilde{y}, \tilde{X}^{(\mathbf{i}_j)} \right\rangle \right)^p
$$

$$
\leq \left(R^2\left(1+\frac{(r_3 p)^{1/2}\sigma_t}{R}\right)+\delta R\right)^{p|\mathcal{U}|}. \tag{20}
$$

Combining Equations (20) and (19) and plugging back into Equation (18):

$$
\mathbb{E}\left(1\{\mathcal{A}_1\} \prod_{j=1}^{r_3} \left\langle \tilde{y}, \tilde{X}^{(\mathbf{i}_j)} \right\rangle \right)^p
$$

$$
= \sum_{\mathcal{U} \subseteq [r_3]} \mathbb{E}\left(1\{\mathcal{A}_1 \cap \mathcal{A}_2(\mathcal{U})\} \prod_{j=1}^{r_3} \left\langle \tilde{y}, \tilde{X}^{(\mathbf{i}_j)} \right\rangle \right)^p
$$

$$
\leq \sum_{\mathcal{U} \subseteq [r_3]} \left(R^2\left(1+\frac{(2r_3 p)^{1/2}\sigma_t}{R}\right)+\delta R\right)^{p|\mathcal{U}|} \left(\nu R\left(\sqrt{2pd}+R+\delta\right)\right)^{(r_3-|\mathcal{U}|)p}
$$

$$
\leq \left(R^2\left(1+\frac{(2r_3 p)^{1/2}\sigma_t}{R}\right)+\delta R+\nu R\left(\sqrt{2pd}+R+\delta\right)\right)^{r_3 p}. \qquad \square
$$

**Lemma E.3** (Replica bounds for the coupled variables). *Suppose we have*

$$
\delta \leq R \wedge \frac{1}{\left(\sqrt{d}+\sqrt{\ln(1/\varepsilon_1)}\right)(\sqrt{2pd}/R+1)^p}.
$$

*Then for $\tilde{y}$ as defined in Section B, let $\tilde{X}^{(i)}$ be independent samples from the posterior distribution $q^{t,\tilde{y}}$. Then, for integers $r_2, r_3, r_4$ that satisfy $r_2 \leq 2(r_3 + r_4) \leq 2k$, under event $\mathcal{A}_1$ defined previously, we have for all tuples of indices $\mathbf{i}, \mathbf{k}, \mathbf{l}$,*

$$\left\| 1\{\mathcal{A}_1\} \frac{(1-\sigma_t^2)^{r_1}}{\sigma_t^{r_2}} \prod_{j=1}^{r_3} \left\langle \tilde{y}, \tilde{X}^{(\mathbf{i}_j)} \right\rangle \prod_{j=1}^{r_4} \left\langle \tilde{X}^{(\mathbf{k}_j)}, \tilde{X}^{(\mathbf{l}_j)} \right\rangle \tilde{X} \right\|_{p,\infty} \lesssim \frac{R}{\sigma_t^{2k}} \left( \frac{R}{\sigma_t} + (kp)^{1/2} \right)^{2k},$$

$$\left\| 1\{\mathcal{A}_1\} \frac{(1-\sigma_t^2)^{r_1}}{\sigma_t^{r_2}} \prod_{j=1}^{r_3} \left\langle \tilde{y}, \tilde{X}^{(\mathbf{i}_j)} \right\rangle \prod_{j=1}^{r_4} \left\langle \tilde{X}^{(\mathbf{k}_j)}, \tilde{X}^{(\mathbf{l}_j)} \right\rangle \tilde{y} \right\|_{p,\infty} \lesssim \frac{R+1}{\sigma_t^{2k}} \left( \frac{R}{\sigma_t} + (kp)^{1/2} \right)^{2k},$$

*where by $\|v\|_{p,\infty}$ we mean the infinity norm of the $p$th moment of the vector $v$.*

*Proof.* Note that given the condition on $\delta$, then in the bound of Equation (15) in Lemma E.2 the second term is dominated by the first term, which implies

$$\left\| 1\{\mathcal{A}_1\} \prod_{j=1}^{r_3} \left\langle \tilde{y}, \tilde{X}^{(\mathbf{i}_j)} \right\rangle \right\|_p \lesssim \left( R^2 \left( 1 + \frac{(r_3 p)^{1/2} \sigma_t}{R} \right) + \delta R \right)^{r_3}$$

$$\lesssim \left( R^2 \left( 1 + \frac{(r_3 p)^{1/2} \sigma_t}{R} \right) \right)^{r_3}.$$

Therefore, from $r_2 \leq 2(r_3 + r_4)$ and using the fact that support of $q_0$ has bounded radius $R$:

$$\left\| 1\{\mathcal{A}_1\} \frac{(1-\sigma_t^2)^{r_1}}{\sigma_t^{r_2}} \prod_{j=1}^{r_3} \left\langle \tilde{y}, \tilde{X}^{(\mathbf{i}_j)} \right\rangle \prod_{j=1}^{r_4} \left\langle \tilde{X}^{(\mathbf{k}_j)}, \tilde{X}^{(\mathbf{l}_j)} \right\rangle \tilde{X} \right\|_{p,\infty}$$

$$\leq \frac{1}{\sigma_t^{r_2}} \left\| 1\{\mathcal{A}_1\} \prod_{j=1}^{r_3} \left\langle \tilde{y}, \tilde{X}^{(\mathbf{i}_j)} \right\rangle \right\|_p R^{2r_4+1}$$

$$\lesssim R \left( \frac{1}{\sigma_t^2} \right)^{r_3+r_4} \left( \frac{R^2}{\sigma_t^2} \right)^{r_4} \left( \frac{R^2}{\sigma_t^2} + \frac{R}{\sigma_t}(r_3 p)^{1/2} \right)^{r_3}$$

$$\leq \frac{R}{\sigma_t^{2k}} \left( \frac{R}{\sigma_t} \right)^{2r_4} \left( \left( \frac{R}{\sigma_t} + (r_3 p)^{1/2} \right)^2 \right)^{r_3}$$

$$\leq \frac{R}{\sigma_t^{2k}} \left( \frac{R}{\sigma_t} + (kp)^{1/2} \right)^{2k}.$$

where we used the fact that $r_3 + r_4 \leq k$. Similarly for the other term, using Cauchy Swartz

$$\left\| 1\{\mathcal{A}_1\} \frac{(1-\sigma_t^2)^{r_1}}{\sigma_t^{r_2}} \prod_{j=1}^{r_3} \left\langle \tilde{y}, \tilde{X}^{(\mathbf{i}_j)} \right\rangle \prod_{j=1}^{r_4} \left\langle \tilde{X}^{(\mathbf{k}_j)}, \tilde{X}^{(\mathbf{l}_j)} \right\rangle y \right\|_{p,\infty}$$

$$\leq \frac{1}{\sigma_t^{r_2}} \left\| \prod_{j=1}^{r_3} \left\langle \tilde{y}, \tilde{X}^{(\mathbf{i}_j)} \right\rangle \right\|_{2p} R^{2r_4} \left( \|\tilde{y} - y\|_{2,\infty} + \|y\|_{2,\infty} \right)$$

$$\leq \frac{1}{\sigma_t^{r_2}} \left\| \prod_{j=1}^{r_3} \left\langle \tilde{y}, \tilde{X}^{(\mathbf{i}_j)} \right\rangle \right\|_{2p} R^{2r_4} \left( R + \sigma_t + \delta \right)$$

$$\lesssim \frac{R+1}{\sigma_t^{2k}} \left( \frac{R}{\sigma_t} + (kp)^{1/2} \right)^{2k}. \qquad \square$$

**Lemma E.4** (High probability control on higher derivatives – fixed time). *In the setting of Lemma E.3 let*

$$g(t) = \tilde{y}_t + \nabla \log q_t(\tilde{y}_t),$$

*where $\tilde{y}(t)$ is defined in Eq. (7). Then, for constant $c_1$ and arbitrary coordinate $i$,*

$$\mathbb{P}\left(\left|\partial_t^k g_i(t)\right| \geq c_1 (60k)^k (R+1) \left(\frac{1}{\sigma_t^2}\right)^k \left(\frac{R}{\sigma_t} + \sqrt{\ln(1/\varepsilon_1)}\right)^{2k}\right) \leq \varepsilon_1.$$

*Proof.* Using Lemmas E.3 and 3.2, there is a constant $c_1$ such that

$$\left\|\partial_t^k g_i(t)\right\|_p = \left\|\partial_t^k (y_{t,i} + \partial_i \log q_t(y_t))\right\|_p$$

$$\leq (c_1/e)(60k)^k (R+1) \left(\frac{1}{\sigma_t^2}\right)^k \left(\frac{R}{\sigma_t} + (kp)^{1/2}\right)^{2k}.$$

Moreover, for fixed $j \leq d$, let $\mathcal{A}_3 \subseteq \mathcal{A}_1$ be the event inside $\mathcal{A}_1$ where

$$\left|\partial_t^k g_i(t)\right| \geq c_1 (60k)^k (R+1) \left(\frac{1}{\sigma_t^2}\right)^k \left(\frac{R}{\sigma_t} + \sqrt{\ln(1/\varepsilon_1)}\right)^{2k}.$$

Taking expectation with respect to $y$

$$\mathbb{E}_y \mathbb{E}_{\{\tilde{X}^{(i)}\}} \left|\partial_t^k g_i(t)\right|^p$$

$$\geq \mathbb{P}(\mathcal{A}_3) \left(c_1 (60k)^k (R+1) \left(\frac{1}{\sigma_t^2}\right)^k \left(\frac{R}{\sigma_t} + \sqrt{\ln(1/\varepsilon_1)}\right)^{2k}\right)^p.$$

Now setting $p = \ln(1/\varepsilon_1)/k$, we get

$$\mathbb{P}(\mathcal{A}_3) \leq \left(\frac{\frac{R}{\sigma_t} + (kp)^{1/2}}{e\frac{R}{\sigma_t} + e\sqrt{\ln(1/\varepsilon_1)}}\right)^{2kp} = \left(\frac{1}{e}\right)^{2\ln(1/\varepsilon_1)} \leq \varepsilon_1. \qquad \square$$

**Lemma E.5.** *In the same setting of Lemma E.4, for*

$$\delta \leq R \wedge \frac{1}{\left(\sqrt{d} + \sqrt{\ln(1/\varepsilon_1)}\right)\left(\sqrt{2\ln(1/\varepsilon)d}/(R\sqrt{k}) + 1\right)^{\ln(1/\varepsilon_1)/k}},$$

*for some constant $c_1$, we have*

$$\mathbb{P}\left(\sup_{\tilde{y}:\|\tilde{y}-y\|\leq\delta} \left|\partial_t^k g_i(t)\right|_{s=0}\right| \geq c_1 (60k)^k (R+1) \left(\frac{1}{\sigma_t^2}\right)^k \left(\frac{R}{\sigma_t} + \sqrt{\ln(1/\varepsilon_1)}\right)^{2k}\right) \leq \varepsilon_1.$$

*Proof.* Note that the $\tilde{y}$ can be an arbitrary mapping of $y$ in the neighborhood $\|y - \tilde{y}\| \leq \delta$. Hence, we pick $\tilde{y}$ to be

$$\tilde{y}(y) \triangleq \operatorname{argmax}_{\tilde{y}:\|\tilde{y}-y\|\leq\delta} \left|\partial_t^k g_i(t)\right|_{t=0}\right|.$$

The result then follows from Lemma E.4. $\qquad \square$

**Lemma E.6.** *Let the distance between the two curves, namely the true and algorithm curves, be upper bounded by*

$$\tilde{\delta} \triangleq R \wedge \frac{1}{\left(\sqrt{d} + \sqrt{\ln(1/\varepsilon_1)}\right)\left(\sqrt{2\ln(1/\varepsilon_1)d}/(R\sqrt{k}) + 1\right)^{\ln(1/\varepsilon_1)/k}}, \tag{21}$$

*that is* $\sup_{t\in[t_0,t_0+h]}\|y(t) - \tilde{y}(t)\| \leq \tilde{\delta}$, *where* $\tilde{y}(t)$ *is defined in Lemma E.5 and* $y$ *is defined analogously with* $y(0) = y$. *Then*

$$\mathbb{P}\left(\sup_{t\in[t_0,t_0+h]}\left|\partial_t^k g_i(t)\right|_{t=0}\right| \geq C_1\right) \leq 2\varepsilon_1.$$

*where*

$$C_1 \triangleq c_1(60k)^k(R+1)\left(\frac{1}{\sigma_t^2}\right)^k\left(\frac{R}{\sigma_t} + \sqrt{\ln\left(\frac{R + \sqrt{d} + \sqrt{\ln(1/\varepsilon_1)}}{\varepsilon_1}\right)}\right)^{2k}.$$

*Proof.* Suppose we take a cover $\mathcal{C}$ of the interval $[t_0, t_0 + h]$ with accuracy $\frac{\tilde{\delta}}{6\left(R+\sqrt{d}+\sqrt{\ln(1/\varepsilon_1)}\right)}$. Then, using Lemma E.5 with a union bound, particularly since $2\tilde{\delta}$ satisfies the condition of this lemma, we get (using $h \leq 1$)

$$\mathbb{P}\left(\sup_{\tilde{y}:\|\tilde{y}-y\|\leq 2\tilde{\delta}, y\in\mathcal{C}}\left|\partial_t^k g_i(t)\right|_{t=0}\right| \geq C_1\right) \leq \varepsilon_1. \tag{22}$$

On the other hand, for every $s \in [t_0, t_0 + h]$, if we assume $\bar{s} \in \mathcal{C}$ be the closest element of the cover to $s$, then from Lemma I.3 and the choice of accuracy of the cover:

$$\|y(\bar{t}) - y(t)\| \leq \tilde{\delta}.$$

Combining this with the distance between the curves and triangle inequality, we get

$$\|\tilde{y}(t) - y(\bar{t})\| \leq 2\tilde{\delta}.$$

Therefore, we showed that every point on the curve $\tilde{y}$ in time interval $[t_0, t_0 + h]$ is close to a point in the cover $\mathcal{C}$. Combining this with Equation (22) implies with probability at least $1 - 2\varepsilon_1$

$$\mathbb{P}\left(\sup_{t\in[t_0,t_0+h]}\left|\partial_t^k g_i(t)\right|_{t=0}\right| \geq C_1\right) \leq 2\varepsilon_1. \qquad \square$$

## F. Proof of Proposition 3.6

In this section we exploit the framework we developed for obtaining a low-degree approximation of the score function on the probability flow ODE using the Picard iteration.

### F.1. Preliminaries

Here we recall the basic setup for Picard iteration with polynomial approximation, as developed in Lee et al. (2018).

**Definition F.1.** *Given vector field* $F(x) : \mathbb{R}^d \to \mathbb{R}^d$, *motivated by the Picard iteration and following Lee et al. (2018), define the operator* $T(x)$ *that acts on a curve* $x(t) : [t_0, t_0 + h] \to \mathbb{R}^d$ *as*

$$T(x)(t) = x(t_0) + \int_{t_0}^{t_0+h} F(x(s))ds.$$

*Furthermore, suppose we have a basis* $\{\phi_j\}_{j=1}^D$ *of smooth one dimensional functions and points* $\{c_j\}_{j=1}^D$ *such that* $1 \leq \forall i, j \leq D$, $\phi_j(c_i) = 0$ *if* $i \neq j$ *and* $\phi_i(c_i) = 1$. *Then, define the approximation* $T_\phi$ *of the* $T$ *operator corresponding to the basis* $\{\phi_j\}_{j=1}^D$ *by*

$$T_\phi(x)(t) = \int_{t_0}^{t_0+h}\sum_{j=1}^D F(x(c_j))\phi_j(s)ds. \tag{23}$$

In particular, we pick $\phi_j$'s to be a basis for one dimensional polynomials of degree less than $D$; we choose them as the Lagrange multiplier polynomials for points $c_j$. This way, the integral in (23) can be computed in closed form.

**Definition F.2.** *We say the basis $\phi = \{\phi_j\}_{j=1}^{D}$ is $\gamma_\phi$ bounded if*

$$\sum_j \left| \int_{t_0}^{t_0+h} \phi_j(s)\,ds \right| \le \gamma_\phi h.$$

Next, we generalize the approach of Lee et al. (2018) and show an important Lipschitz property of $T_\phi$ with respect to the supremum norm $\|.\|$, when the vector field $F$ is close to a Lipschitz one.

**Lemma F.3.** *Suppose the basis $\phi$ is $\gamma_\phi$ bounded, and the vector field $F_s$ is $\tilde{L}$ Lipschitz. Then, for arbitrary $x, y \in \mathcal{C}([t_0, t_0 + h], \mathbb{R}^d)$ with $x(t_0) = y(t_0)$,*

$$\|T_\phi(x) - T_\phi(y)\|_{[t_0, t_0+h]} \le \tilde{L}\|x - y\|_{[t_0, t_0+h]}\gamma_\phi h,$$

$$\|T_\phi^{\circ \ell}(x) - T_\phi^{\circ \ell}(y)\|_{[t_0, t_0+h]} \le \left(\tilde{L}\gamma_\phi h\right)^{\ell} \|x - y\|_{[t_0, t_0+h]}. \tag{24}$$

*Proof.* For every $1 \le s \le h$, using $\tilde{L}$ Lipschitz property of our estimate for the score function,

$$\|F_t(x(t)) - F_t(y(t))\| \le \tilde{L}\|x(t) - y(t)\|$$
$$\le \tilde{L}\|x - y\|_{[t_0, t_0+h]}.$$

Hence

$$\|T_\phi(x) - T_\phi(y)\|_{[t_0, t_0+h]} = \sup_{t_0 \le h' \le t_0+h} \left\| \int_{t_0}^{t_0+h'} \sum_{j=1}^{D} \left( F_{c_j}(x(c_j)) - F_{c_j}(y(c_j)) \right) \phi_j(s)ds \right\|$$

$$\le \sup_{1 \le j \le D} \left\| F_{c_j}(x(c_j)) - F_{c_j}(y(c_j)) \right\|_\infty \sum_j \left| \int_{t_0}^{t_0+h} \phi_j(s)ds \right|$$

$$\le \tilde{L}\|x - y\|_{[t_0, t_0+h]}\gamma_\phi h.$$

the second line in (24) follows from applying the first line in (24) $\ell$ times. $\qquad \square$

**Definition F.4.** *(Low-degree vector field) Let $y_t$ be the solution to the ODE $\dot{y}_t = F_t^*(y_t)$. We say the vector field $F^*$ is low-degree along $y_t$ if it accepts the following low degree approximation:*

$$\|F^* \circ y - P_{\le D}(F^* \circ y)\|_{[t_0, t_0+h]} \le \varepsilon_{\mathsf{ld}},$$

*where the curve $P_{\le D}(F^* \circ y)$ is an approximation of $F^* \circ y$ whose coordinates are degree at most $D$ polynomials in the time variable $t \in [t_0, t_0 + h]$.*

**Lemma F.5.** *For $\dot{y}_t = F_t^*(y_t)$, given that $F^*$ is a low-degree vector field based on Definition F.4, we have*

$$\left\| T_\phi^{\circ m} y - y \right\| \le \frac{\left(\tilde{L}\gamma_\phi h\right)^m - 1}{\left(\tilde{L}\gamma_\phi h\right) - 1} (\varepsilon_{\mathsf{ld}} + \max_{j=1}^{D} \|F(y(c_j)) - F^*(y(c_j))\|)\,(1 + \gamma_\phi)\,h.$$

*Proof.* We can write $T_\phi(y) = y_{t_0} + S_\phi(F \circ y)$, where for arbitrary curve $z \in \mathcal{C}([t_0, t_0 + h], \mathbb{R}^d)$ define

$$S_\phi(z)(.) = \sum_{j=1}^{D} z(c_j) \int_{t_0}^{\cdot} \phi_j(s)ds.$$

Note that because $P_{\le D}(F^* \circ y)$ is degree at most $D$ and $\phi_j$'s are the Lagrange multiplier polynomials at the Chebyshev points $(c_i)_{j=1}^{D}$, then

$$P_{\le D}(F^* \circ y) = \sum_{j=1}^{D} P_{\le D}(F^* \circ y)(c_j)\phi_j,$$

which from the definition of $S_\phi(z)$, implies

$$\int_{t_0}^{\cdot} P_{\leq D} \left( F^* \circ y \right)(s) ds = S_\phi \left( P_{\leq D} \left( F^* \circ y \right) \right).$$

Now combining this with Lemma F.3 and using the low-degree Definition F.4

$$\left\| S_\phi(F \circ y) - \int_{t_0}^{\cdot} P_{\leq D} \left( F^* \circ y \right)(s) ds \right\|_{[t_0, t_0+h]}$$

$$= \left\| S_\phi(F \circ y) - S_\phi \left( P_{\leq D} \left( F^* \circ y \right) \right) \right\|_{[t_0, t_0+h]}$$

$$= \left\| \sum_{j=1}^{D} \left( F \circ y \Big|_{c_j} - P_{\leq D} \left( F^* \circ y \right) \Big|_{c_j} \right) \left( y_{t_0} + \int_{t_0}^{\cdot} \phi_j(s) ds \right) \right\|_{[t_0, t_0+h]}$$

$$\leq \left( \max_{j=1}^{D} \| F^*(y(c_j)) - P_{\leq D} \left( F^*(y(c_j)) \right) \| + \| F(y(c_j)) - F^*(y(c_j)) \| \right) \sum_{j=1}^{D} \left\| y_{t_0} + \int_{t_0}^{\cdot} \phi_j(s) ds \right\|_{[t_0, t_0+h]}$$

$$\leq \left( \varepsilon_{\mathsf{ld}} + \max_{j=1}^{D} \| F(y(c_j)) - F^*(y(c_j)) \| \right) \sum_{j=1}^{D} \left\| y_{t_0} + \int_{t_0}^{\cdot} \phi_j(s) ds \right\|_{[t_0, t_0+h]}$$

$$\leq \left( \varepsilon_{\mathsf{ld}} + \max_{j=1}^{D} \| F(y(c_j)) - F^*(y(c_j)) \| \right) \gamma_\phi h. \tag{25}$$

But from the definition of $\varepsilon_{\mathsf{ld}}$,

$$\| P_{\leq D} \left( F^* \circ y \right) - F^* \circ y \|_{[t_0, t_0+h]} \leq \varepsilon_{\mathsf{ld}},$$

which using the identity $y - y_{t_0} = \int_{t_0}^{\cdot} F \circ y(s) ds$ implies

$$\left\| \int_{t_0}^{\cdot} P_{\leq D} \left( F^* \circ y \right)(s) ds - (y - y_{t_0}) \right\|_{[t_0, t_0+h]} = \left\| \int_{t_0}^{\cdot} P_{\leq D} \left( F^* \circ y \right)(s) ds - \int_{t_0}^{\cdot} F^* \circ y(s) ds \right\|_{[t_0, t_0+h]}$$

$$\leq \varepsilon_{\mathsf{ld}} h \tag{26}$$

Combining (25) and (26)

$$\| T_\phi(y) - y \|_{[t_0, t_0+h]} \leq \left\| T_\phi(y) - y_{y_{t_0}} - \int_{t_0}^{\cdot} P_{\leq D} \left( F^* \circ y \right) ds \right\|_{[t_0, t_0+h]}$$

$$+ \left\| \int_{t_0}^{\cdot} P_{\leq D} \left( F^* \circ y \right) ds - (y - y_{t_0}) \right\|_{[t_0, t_0+h]}$$

$$\leq \left( \varepsilon_{\mathsf{ld}} + \max_{j=1}^{D} \| F(y(c_j)) - F^*(y(c_j)) \| \right) (1 + \gamma_\phi) h.$$

Now applying Lemma F.3 $i - 1$ times

$$\left\| T_\phi^{\circ i}(y) - T_\phi^{\circ(i-1)}(y) \right\| \leq \left( \tilde{L} \gamma_\phi h \right)^{i-1} \| T_\phi(y) - y \|_{[0,h]}$$

$$\leq \left( \tilde{L} \gamma_\phi h \right)^{i-1} \left( \varepsilon_{\mathsf{ld}} + \max_{j=1}^{D} \| F(y(c_j)) - F^*(y(c_j)) \| \right) (1 + \gamma_\phi) h \tag{27}$$

Summing (27) for $i = 1, \ldots, m$:

$$\left\| T_\phi^{\circ m}(y) - y \right\| \leq \sum_{i=1}^{m} \left\| T_\phi^{\circ i} y - T_\phi^{\circ(i-1)} y \right\|$$

$$\leq \frac{\left( \tilde{L} \gamma_\phi h \right)^m - 1}{\left( \tilde{L} \gamma_\phi h \right) - 1} \left( \varepsilon_{\mathsf{ld}} + \max_{j=1}^{D} \| F(y(c_j)) - F^*(y(c_j)) \| \right) (1 + \gamma_\phi) h. \qquad \square$$

**Corollary F.6** (Effect of approximate Picard iterations). *For the exact probability flow ODE $\dot{y}_t = F_t^*(y_t)$, given that $F^*$ is a low-degree vector field based on Definition F.4 and given $\tilde{L}\gamma_\phi h \leq \frac{1}{2}$, for arbitrary curve $x \in \mathcal{C}([t_0, t_0 + h], \mathbb{R}^d)$ we have*

$$\left\|T_\phi^{\circ m}(y) - y\right\|_{[t_0, t_0+h]} \leq 2(\varepsilon_{\mathsf{ld}} + \max_{j=1}^{D} \|F(y(c_j)) - F^*(y(c_j))\|)(1 + \gamma_\phi) h$$

$$\left\|T_\phi^{\circ m}(x) - T_\phi^{\circ m}(y)\right\|_{[t_0, t_0+h]} \leq \frac{1}{2^m} \|x - y\|_{[t_0, t_0+h]}.$$

*Proof.* This follows directly from Lemmas F.3 and F.5. $\qquad\square$

**Lemma F.7.** *Recall the definition of $F$ and $F^*$ from Section B. For initial point $\|\bar{y}_{t_0} - y_{t_0}\| \leq \varepsilon_p$, define $y_s$, $\tilde{y}_s$, and $\bar{y}_s$ for $t_0 \leq s \leq t_0 + h$ as*

$$\dot{y}_s = F_s^*(y_s),$$
$$\dot{\tilde{y}}_s = F_s^*(\tilde{y}_s),$$
$$\bar{y}(s) = \bar{y}_{t_0} + (s - t_0)F_{t_0}(\bar{y}_{t_0}).$$

*Then, under the low degree Assumption F.4 for the curve $\tilde{y}$ in the time interval $[t_0, t_0+h]$, and picking step size $h = O(\frac{1}{1+R^2})$ and assuming $\tilde{L} \leq \frac{1}{2}$ and $\sigma_t \geq 1$,*

$$\left\|y - T_\phi^{\circ m}(\bar{y})\right\|_{[t_0, t_0+h]} \leq 2\varepsilon_p(1 + 2(1 + \gamma_\phi)h) + (\varepsilon_{\mathsf{ld}} + \max_{j=1}^{D} \|F(y(c_j)) - F^*(y(c_j))\|)(1 + \gamma_\phi) h$$

$$+ \frac{h}{2^m}\left(\tilde{L}\varepsilon_p + \varepsilon_{\mathsf{err}} + \sqrt{d}R(R^2 + \ln(1/\varepsilon_1)) + (2 + 4R^2)\varepsilon_p\right).$$

*Proof.* First, since we have the assumption $T - t_0 - h \geq 1$, we have $L_s \leq (2 + 4R^2), \forall s \in [t_0, t_0 + h]$. Hence, since we know $\gamma_\phi = O(1)$ for the Chebyshev basis, the assumption $h \leq O(\frac{1}{1+R^2})$ satisfies the precondition of Corollary 3.6 on $h$. On the other hand, from Lemma I.2, again from the condition $h = O(\frac{1}{1+R^2})$, we get $\|y - \tilde{y}\|_{[t_0, t_0+h]} \leq 2\varepsilon_p$. Now combining this with Corollary 3.6:

$$\left\|y - T_\phi^{\circ m}(\bar{y})\right\|_{[t_0, t_0+h]} \leq \|y - \tilde{y}\|_{[t_0, t_0+h]} + \left\|\tilde{y} - T_\phi^{\circ m}(\tilde{y})\right\|_{[t_0, t_0+h]} + \left\|T_\phi^{\circ m}(\bar{y}) - T_\phi^{\circ m}(\tilde{y})\right\|_{[t_0, t_0+h]}$$

$$\leq 2\varepsilon_p + 2(\varepsilon_{\mathsf{ld}} + \max_{j=1}^{D} \|F(\tilde{y}(c_j)) - F^*(\tilde{y}(c_j))\|)(1 + \gamma_\phi) h + \frac{1}{2^m}\|\bar{y} - \tilde{y}\|_{[t_0, t_0+h]}$$

$$\leq 2\varepsilon_p + 2(\varepsilon_{\mathsf{ld}} + 2\|y - \tilde{y}\|_{[t_0, t_0+h]} + \max_{j=1}^{D} \|F(y(c_j)) - F^*(y(c_j))\|)(1 + \gamma_\phi) h$$

$$+ \frac{1}{2^m}\|\bar{y} - \tilde{y}\|_{[t_0, t_0+h]}$$

$$\leq 2\varepsilon_p(1 + 2(1 + \gamma_\phi)h) + 2(\varepsilon_{\mathsf{ld}} + \max_{j=1}^{D} \|F(y(c_j)) - F^*(y(c_j))\|)(1 + \gamma_\phi) h$$

$$+ \frac{1}{2^m}\|\bar{y} - \tilde{y}\|_{[t_0, t_0+h]} \tag{28}$$

But from Lemma I.3, with probability at least $1 - \varepsilon_1$, for all $t_0 \leq s \leq t_0 + h$

$$\|F_s^*(\tilde{y}_s)\| \leq 6\left(R + \sqrt{d} + \sqrt{\ln(1/\varepsilon_1)}\right). \tag{29}$$

Hence, using Lemma I.1 and I.2,

$$\frac{d}{ds}\|\bar{y}_s - \tilde{y}_s\| \leq \frac{\langle F_{t_0}(\tilde{y}_{t_0}) - F_s^*(\tilde{y}_s), \bar{y}_t - y_t\rangle}{\|\bar{y}_t - y_t\|}$$

$$\leq \|F_{t_0}(\tilde{y}_{t_0}) - F_t^*(\tilde{y}_t)\|$$

$$\leq \|F_{t_0}(\tilde{y}_{t_0}) - F_{t_0}(y_{t_0})\| + \|F_{t_0}(y_{t_0}) - F_{t_0}^*(y_{t_0})\|$$

$$+ \|F_{t_0}^*(y_{t_0}) - F_t^*(y_t)\| + \|F_t^*(y_t) - F_t^*(\tilde{y}_t)\|$$

$$\leq \tilde{L}\varepsilon_p + \varepsilon_{\mathsf{err}} + \sqrt{d}R(R^2 + \ln(1/\varepsilon_1)) + (2 + 4R^2)e^{\frac{5R^2}{\sigma_t^2}h}\varepsilon_p.$$

which implies

$$\|\bar{y} - \tilde{y}\|_{[t_0, t_0+h]} \leq h\left(\left\|F_{t_0}(\tilde{y}_{t_0}) - F_{t_0}^*(\tilde{y}_{t_0})\right\| + 12\left(R + \sqrt{d} + \sqrt{\ln(1/\varepsilon_1)}\right)\right).$$

Plugging this back into (28)

$$\left\|y - T_\phi^{\circ m}(\bar{y})\right\| \leq 2\varepsilon_p(1 + 2(1+\gamma_\phi)h) + \left(\varepsilon_{\mathsf{ld}} + \max_{j=1}^{D} \|F(y(c_j)) - F^*(y(c_j))\|\right)(1+\gamma_\phi)h$$

$$+ \frac{h}{2^m}\left(\tilde{L}\varepsilon_p + \varepsilon_{\mathsf{err}} + \sqrt{d}R(R^2 + \ln(1/\varepsilon_1)) + (2+4R^2)e^{\frac{5R^2}{\sigma_t^2}h}\varepsilon_p\right).$$

Using the assumption $h = O(\frac{1}{1+R^2})$ completes the proof. $\qquad\square$

## G. Proof of Theorem 3.7

In this section, we put everything together to show our main end-to-end result, namely how to sample from the target distribution which is $\tilde{O}(\varepsilon_{\mathsf{err}}R^2)$-close in TV to a distribution which is $\tilde{O}(\varepsilon_{\mathsf{err}})$-close in $W_2$ to the true distribution $q$.

**Theorem G.1** (Formal version of Theorem 3.7). *For error parameters $\varepsilon_1, \varepsilon_{\mathsf{err}} > 0$ which satisfy the bounds $\varepsilon_{\mathsf{err}} \leq \varepsilon_1/\ln(d/R)$ and $\varepsilon_{\mathsf{err}} \leq \left((R/\sigma) \wedge (1/\sqrt{d})\right)/(\ln(\sigma/R) + \ln(d))^2$, given step size $h = \frac{\gamma}{k(1+(R/\sigma)^2)(\ln(1/\varepsilon_1)+\ln(d))}$ for small enough constant $\gamma$, where $k \triangleq \ln(1/\varepsilon_{\mathsf{err}})$ is the degree of the polynomial approximation, with probability at least*

$$1 - O\left(\varepsilon_1 T(R/\sigma)^2 \ln(1/\varepsilon_{\mathsf{err}})(\ln(1/\varepsilon) + \ln(d))\right),$$

*for $T = \ln(R/\sigma) + \ln(d) + \ln(1/\varepsilon_{\mathsf{err}})$, Algorithm 2 outputs a sample whose distribution is close in 2-Wasserstein distance to the target measure $q$ to within error*

$$O(Tk\varepsilon_{\mathsf{err}}\log(1/\varepsilon_1)) = O\left((\ln(R/\sigma) + \ln(d) + \ln(1/\varepsilon_{\mathsf{err}}))\ln(1/\varepsilon_{\mathsf{err}})\ln(k/\varepsilon_1)\varepsilon_{\mathsf{err}}\right),$$

*in*

$$O\left(T(R/\sigma)^2 \ln(1/\varepsilon_{\mathsf{err}})(\ln(1/\varepsilon_1) + \ln(d))\right)$$

*number of rounds and $m = \ln(R/\sigma) + \ln(d) + \ln\ln(1/\varepsilon_1) + \ln(1/\varepsilon_{\mathsf{err}}) + \ln(\tilde{L})$ number of Picard iterations in each round.*

*Proof.* We can scale the distribution, sample from the scaled distribution, then scale back. Note that this scaling procedure does not change the ratio $R/\sigma$. Therefore, without loss of generality, we assume $\sigma_0 = \sigma = 1$, since our bounds only depend on the quantity $R/\sigma$. We run the forward process up to time

$$T := \Theta(\ln(R) + \ln(d) + \ln(1/\varepsilon_{\mathsf{err}})). \tag{30}$$

Now we prove inductively that $\left\|y - T_\phi^{\circ m}(\bar{y})\right\|_{[t^{(i)}, t^{(i)}+h]} = O((t^{(i)} + h)k\varepsilon_{\mathsf{err}}\log(k/\varepsilon_1))$ for all $i$. First, note that we can bound the Wasserstein distance of the target distribution $q_{\mathsf{pre}}$ and $\gamma^d$ as

$$W_2(q_{\mathsf{pre}}, \gamma^d) \leq W_2(q_{\mathsf{pre}}, \delta_{\{0\}}) + W_2(\delta_{\{0\}}, \gamma^d),$$

where $\delta_{\{0\}}$ is the point mass at the origin. But

$$W_2(q_{\mathsf{pre}}, \delta_{\{0\}}) = \mathbb{E}_{q_{\mathsf{base}}}\|Y\|^2 \leq R,$$
$$W_2(\gamma^d, \delta_{\{0\}}) = \sqrt{d}.$$

Therefore

$$W_2(q_{\mathsf{pre}}, \gamma^d) \leq R + \sqrt{d}.$$

Now from the Wasserstein contraction property of the OU process and the choice of $T$ in (30), we get

$$W_2(q_T, \gamma^d) \leq \left(R + \sqrt{d}\right)e^{-T} = O(\varepsilon_{\mathsf{err}}).$$

To prove the step of induction for the interval $[t^{(i)}, t^{(i)} + h] = [t_0, t_0 + h]$, we know from the hypothesis of induction that for all previous intervals $(t^{(j)}, t^{(j)} + h)$ for $j < i$:

$$\left\|y - T^{\circ m}_{\phi}(\bar{y})\right\|_{[t^{(j)}, t^{(j)} + h]} \leq O((t^{(j)} + h)k\varepsilon_{\mathsf{err}} \log(k/\varepsilon_1)),$$

which from the definition of the iterates of the algorithm, i.e. $\hat{y} = T^{\circ m}_{\phi}(\bar{y})$ implies

$$\|y - \hat{y}\|_{[0, t_0]} \leq O(t_0 k\varepsilon_{\mathsf{err}} \log(k/\varepsilon_1)).$$

Using Lemma I.2, this further implies, given the choice of $h$,

$$\|\tilde{y} - y\|_{[t_0, t_0 + h]} = O(t_0 k\varepsilon_{\mathsf{err}} \log(k/\varepsilon_1)).$$

Hence, we can use $\varepsilon_p = O(t_0 k\varepsilon_{\mathsf{err}} \log(k/\varepsilon_1))$ in Lemma F.7.

Now given the assumptions $\varepsilon_{\mathsf{err}} \leq (1/\ln(d/R))\varepsilon_1$ and $\varepsilon_{\mathsf{err}} \leq \left(R \wedge (1/\sqrt{d})\right)/(\ln(1/R) + \ln(d))^2$, and since $\sigma_t \geq \sigma_0 \geq 1$ it is easy to check that $\tilde{\delta} = O(t_0 k\varepsilon_{\mathsf{err}} \log(k/\varepsilon_1))$ satisfies the assumption of Lemma E.6.

Hence, we can apply Lemma E.6 with

$$k = \ln(1/\varepsilon_{\mathsf{err}}),$$

using the fact that $\sigma_t \geq \frac{1}{2}$ and $R \geq 1$ there is constant $c_2$ such that for the interval $[t_0, t_0 + h] := [t^{(i)}, t^{(i)} + h]$,

$$\mathbb{P}\left(\sup_{s \in [t_0, t_0 + h]} \left|\partial_s^k g_i^{(\tilde{y}(s))}(s)\right|_{s=0}\right| \geq \left(c_2 k^2 R^2 \left(\ln(1/\varepsilon_1) + \ln(d)\right)\right)^k\right) \leq 2\varepsilon_1.$$

Moreover, using this with Fact C.1 and by picking step size

$$h = \frac{\gamma}{k(1 + R^2)(\ln(1/\varepsilon_1) + \ln(d))}$$

for small enough constant $\gamma$, we get with probability at least $1 - 2\varepsilon_1$

$$\|g - P_{\leq k}g\|_{[t_0, t_0 + h]} \leq \left(hc_2 ekR^2 \left(\ln(1/\varepsilon_1) + \ln(d)\right)\right)^k \leq \varepsilon_{\mathsf{err}}. \tag{31}$$

Therefore, we can now use Lemma F.7 with

$$m := \ln(R) + \ln(d) + \ln\ln(1/\varepsilon_1) + \ln(1/\varepsilon_{\mathsf{err}}) + \ln(\tilde{L})$$

number of Picard iterations; using the fact that $\gamma_{\phi} = O(1)$ for the Chebyshev basis $(\phi_0, \ldots, \phi_k)$ for the polynomials defined in Section 2.3, setting $\varepsilon_{\mathsf{Id}} = \varepsilon_{\mathsf{err}}$ in Assumption F.4 based on Equation (31), and using Assumption 2, we get

$$
\begin{aligned}
\left\|y - T^{\circ m}_{\phi}(\bar{y})\right\|_{[t_0, t_0 + h]} = {} & O\left(\varepsilon_p + h\left(\varepsilon_{\mathsf{err}} + \sum_{j=1}^{k} \|F(y(c_j)) - F^*(y(c_j))\|\right)\right) \\
& + \frac{h}{2^m} O\left(\tilde{L}\varepsilon_p + \varepsilon_{\mathsf{err}} + \sqrt{d}R(R^2 + \ln(1/\varepsilon_1)) + (2 + 4R^2)\varepsilon_p\right) \\
\leq {} & O(\varepsilon_p + hk\varepsilon_{\mathsf{err}} \log(k/\varepsilon_1)) \\
& + \frac{h}{2^m} O\left(\tilde{L}\varepsilon_p + \varepsilon_{\mathsf{err}} + \sqrt{d}R(R^2 + \ln(1/\varepsilon_1)) + (2 + 4R^2)\varepsilon_p\right)., \tag{32}
\end{aligned}
$$

where the last line follows from the choice of $m$.

Now using the fact that $h = O(\frac{1}{1+R^2})$, we have

$$\frac{h}{2^m}(2 + 4R^2)\varepsilon_p = O(\frac{1}{2^m} Tk\varepsilon_{\mathsf{err}} \log(1/\varepsilon_1)) = O(\frac{T/h}{2^m} hk\varepsilon_{\mathsf{err}} \log(1/\varepsilon_1)) = O(hk\varepsilon_{\mathsf{err}} \log(k/\varepsilon_1)),$$

where we used the fact that

$$T/h = O\left((1 + R^2)\ln(1/\varepsilon_{\text{err}})^2\left(\ln(1/\varepsilon_1) + \ln(d)\right)\right) = O(2^m).$$

Similarly, it is not hard to check that with the choice of $m$ we have

$$\frac{h}{2^m}\left(\tilde{L}\varepsilon_p + \varepsilon_{\text{err}} + \sqrt{d}R(R^2 + \ln(1/\varepsilon_1))\right) = O(hk\varepsilon_{\text{err}}\log(k/\varepsilon_1))$$

Therefore, we can upper bound (32) as

$$\mathbb{E}\left\|y - T_\phi^{\circ m}(\bar{y})\right\|_{[t_0, t_0 + h]} \leq O(\varepsilon_p) + O(hk\varepsilon_{\text{err}}\log(k/\varepsilon_1)) = O((t_0 + h)k\varepsilon_{\text{err}}\log(k/\varepsilon_1)).$$

which proves the step of induction. Therefore, overall we showed

$$\|y - \hat{y}\|_{[0,T]} = O(Tk\varepsilon_{\text{err}}\log(1/\varepsilon_1)) = O\left(\left(\ln(R) + \ln(d) + \ln(1/\varepsilon_{\text{err}})\right)\ln(1/\varepsilon_{\text{err}})\log(k/\varepsilon_1)\varepsilon_{\text{err}}\right),$$

after $O\left(TR^2\ln(1/\varepsilon_{\text{err}})\left(\ln(1/\varepsilon_1) + \ln(d)\right)\right)$ number of rounds, where in each round we apply the Picard iteration for $m = \ln(R) + \ln(d) + \ln\ln(1/\varepsilon_1) + \ln(1/\varepsilon_{\text{err}}) + \ln(\tilde{L})$ number of times. Note that this Wasserstein guarantee holds only with probability at least $1 - O(\varepsilon_1 T/h) = 1 - O\left(\varepsilon_1 TR^2\ln(1/\varepsilon_{\text{err}})(\ln(1/\varepsilon) + \ln(d))\right)$ after applying a union bound. $\square$

## H. Proof of Corollary 3.9

In this section we show how to use underdamped Langevin Monte Carlo to upgrade the Wasserstein bound achieved by our sampler into a total variation bound, thus proving Corollary 3.9. The pseudocode for this is provided in Algorithm 3, where we propose a slight modification of Algorithm 2 that relies on running the underdamped Langevin Monte Carlo algorithm (see Appendix H.1) for a short period of time at the end of Algorithm 2. This step is often referred to as a *corrector step* in the diffusion model literature. The resulting Algorithm 3 admits a TV guarantee, stated in Corollary 3.9.

---

**Algorithm 3:** CORRECTEDCOLLOCATIONDIFFUSION($(s_t), \varepsilon$)

    **Input:** Score estimates $s_t$ satisfying Assumption 2, target error $0 < \varepsilon < 1$
    **Output:** Sample from a distribution $\hat{q}$ satisfying $\text{TV}(\hat{q}, q) \leq \varepsilon$
**1** $\eta \leftarrow \Theta(\min(\varepsilon^{5/3}/(L^{1/4}d^{1/2}), \varepsilon\sigma^2/R^2))$, where $L \leq \text{poly}(d/\eta)$ is an upper bound on the Lipschitz constant of $q$.
**2** $x \leftarrow$ COLLOCATIONDIFFUSION($(s_t), \varepsilon$)
**3** Run underdamped Langevin Monte Carlo (see Appendix H.1) with friction parameter $\Theta(\sqrt{L})$ and step size
    $h = \varepsilon^{2/3}/(d^{1/3}M(\eta)^{1/3}L^{1/2})$ for $M(\eta)$ steps, where $M(\eta)$ denotes the number of steps used to run
    COLLOCATIONDIFFUSION in the previous step. Let the resulting sample be $x'$.
**4** **return** $x'$

---

### H.1. Underdamped Langevin Monte Carlo

In this section, for the sake of completeness, we briefly review underdamped Langevin Monte Carlo, which is only used in this final phase of our algorithm to convert from Wasserstein closeness to TV closeness.

Given an estimate $s$ of the log-density of a distribution $q$, and a *friction parameter* $\gamma$, underdamped Langevin Monte Carlo with step size $h$ and score estimate $s$ is given by

$$\begin{aligned}
\mathrm{d}x_t &= v_t\,\mathrm{d}t \\
\mathrm{d}v_t &= (s(x_{\lfloor t/h \rfloor h}) - \gamma v_t)\,\mathrm{d}t + \sqrt{2\gamma}\,\mathrm{d}B_t,
\end{aligned}$$

where $B_t$ is a standard Brownian motion.

We will use the following result of Chen et al. (2024c), which is a consequence of the short-time regularization of Guillin & Wang (2012):

**Theorem H.1** (Theorem A.5 of Chen et al. (2024c), restated). *Let $q \propto e^{-H}$ be a distribution over $\mathbb{R}^d$ for which $\nabla H$ is $L$-Lipschitz. Let $p$ be an arbitrary distribution over $\mathbb{R}^d$. Suppose $s : \mathbb{R}^d \to \mathbb{R}^d$ satisfies $\|s - \nabla H\|_{L_2(q)}^2 \le \varepsilon_{\mathrm{sc}}^2$. Let $T \lesssim 1/\sqrt{L}$.*

*If $p_N$ denotes the distribution given by running underdamped Langevin Monte Carlo initialized at $p$ for $T/h$ steps and step size $h$ with friction parameter $\Theta(\sqrt{L})$, then*

$$\mathrm{TV}(p_N, q) \lesssim \frac{W_2(p, q)}{L^{1/4} T^{3/2}} + \frac{\varepsilon_{\mathrm{sc}} T^{1/2}}{L^{1/4}} + L^{3/4} T^{1/2} d^{1/2} h. \tag{33}$$

## H.2. Completing the proof

*A priori* it might appear that the third term in Eq. (33) forces a choice of $h = O(d^{-1/2}\varepsilon)$, translating to an iteration complexity of $\Omega(d^{1/2}/\varepsilon)$. Here we use an idea of Gupta et al. (2024a): because in our application of Theorem H.1, $W_2(p, q)$ can be made quite small, we can take $T$ to be small to get a much better bound on the iteration complexity $T/h$.

*Proof of Corollary 3.9.* For $\eta$ to be tuned, let $M(\eta) = (R/\sigma)^2 \cdot \mathrm{polylog}(1/\varepsilon, d, R, \tilde{L})$ denote the number of iterations of Algorithm 2 needed to achieve Wasserstein error $\eta$ in Theorem 3.7. We will take $h = \varepsilon^{2/3}/(d^{1/3} M(\eta)^{1/3} L^{1/2})$ and $T = M(\eta)h$ in Theorem H.1 to conclude that, starting from the distribution given by Algorithm 2, if we run underdamped Langevin Monte Carlo for $M(\eta)$ iterations, we will produce a distribution $p_N$ for which

$$\mathrm{TV}(p_N, q) \lesssim \frac{\eta L^{1/4} d^{1/2}}{M(\eta)^{2/3} \varepsilon^{2/3}} + \frac{\varepsilon_{\mathrm{sc}} M(\eta)^{1/3} \varepsilon^{1/3}}{L^{1/2} d^{1/6}} + \varepsilon, \tag{34}$$

conditioned on the event that Algorithm 2 succeeds in achieving Wasserstein error $\eta$. The latter happens with probability $1 - \eta(R/\sigma)^2$, so the overall TV error of Algorithm 3 is given by Eq. (34) plus $\eta(R/\sigma)^2$.

If we take $\eta = \min(\varepsilon^{5/3}/(L^{1/4} d^{1/2}), \varepsilon\sigma^2/R^2)$, then $\eta(R/\sigma)^2 \le \varepsilon$, $M(\eta) \le \tilde{O}(R/\sigma)^2 \cdot \mathrm{polylog}(1/\varepsilon, d, \tilde{L}, L)$, and the first term on the right-hand side is bounded by $\varepsilon$. By the assumption that $\varepsilon_{\mathrm{sc}} \le \tilde{O}(\frac{\varepsilon^{2/3} L^{1/2} d^{1/6}}{(R/\sigma)^{2/3}})$, the second term on the right-hand side is bounded by $\varepsilon$. By replacing $\varepsilon$ with $c\varepsilon$ for sufficiently small constant $c$ in the above, we obtain the claimed bound. $\square$

# I. Upper estimates on the movement of the probability flow ODE

In this section we prove some useful properties of the score function. First, we bound the smoothness of the score function $\nabla \log q_t$.

**Lemma I.1.** *(Bounding the operator norm of the true score) For $t \ge 1$, we have*

$$\|\nabla (y_t + \nabla \log q_t(y_t))\| \le e^{-(T-t)} \left(2 + 4R^2\right),$$
$$\left\|\nabla^2 \log q_t(y_t)\right\|_{\mathrm{op}} \le 2 + 4e^{-(T-t)} R^2,$$

*and for $2(T - t) < 1$,*

$$\|\nabla (y_t + \nabla \log q_t(y_t))\| \le \frac{5R^2}{(T-t)^2}.$$

*Proof.* It is easy to check the Hessian of $\log q_t$ can be written in the following form:

$$-\nabla^2 \log q_t(y) = -\frac{1}{\sigma_t^4} \mathrm{Cov}(q^{t,y}) + \frac{1}{\sigma_t^2} I.$$

But using the radius $R$ assumption on the support of $q_{\mathrm{pre}}$, we can bound the covariance as

$$\mathrm{Cov}(q^{t,y}) \le \frac{1 - \sigma_t^2}{\sigma_t^4} R^2 I.$$

Therefore

$$-\nabla\left(y_t + \nabla \log q_t(y_t)\right) = \frac{1 - \sigma_t^2}{\sigma_t^2} I - \frac{1 - \sigma_t^2}{\sigma_t^4} R^2 I. \tag{35}$$

Now for $t \geq 1$, we have $\sigma_t^2 = (1 - e^{-2(T-t)}) \geq 1 - 1/e^2 \geq 0.5$. Therefore, for $t \geq 1$,

$$\|\nabla\left(y_t + \nabla \log q_t(y_t)\right)\| \leq (1 - \sigma_t^2)\left(2 + 4R^2\right) = e^{-2(T-t)}\left(2 + 4R^2\right).$$

Similarly

$$\left\|\nabla^2 \log q_t(y_t)\right\| \leq 2 + (1 - \sigma_t^2)\left(4R^2\right) = 2 + 4e^{-2(T-t)}R^2.$$

On the other hand, for $s = 2(T-t) < 1$, using $e^{-s} \leq 1 - s + \frac{s^2}{2} \leq 1 - \frac{s}{2}$ we have $\sigma_t^2 = 1 - e^{-(T-t)} \geq T - t$. Then from the assumption $R \geq 1$ and Equation (35), we have

$$\|\nabla\left(y_t + \nabla \log q_t(y_t)\right)\| \leq \left(\frac{2}{T-t} + \frac{4R^2}{(T-t)^2}\right) \leq \frac{5R^2}{(T-t)^2}. \qquad \square$$

We can then use this bound on the smoothness to control the extent to which two processes evolving according to the same probability flow ODE diverge over time:

**Lemma I.2.** *(Distance between true ODE solutions starting from close points) For $y_t, \tilde{y}_t$ that evolve according to probability flow ODE, i.e.,*

$$\frac{d}{dt} y_t = y_t + \nabla \log q_t(y_t)$$

$$\frac{d}{dt} \tilde{y}_t = \tilde{y}_t + \nabla \log q_t(\tilde{y}_t),$$

*with initial condition satisfying $\|y_{t_0} - \tilde{y}_{t_0}\| \leq \varepsilon$, then, for time window $s$ such that $T - (t_0 + s) \geq 1$, we have*

$$\|y_{t_0+s} - \tilde{y}_{t_0+s}\| \leq e^{\frac{5R^2}{\sigma_t^2}s}\varepsilon$$

*Proof.* Note that for $t_0 \leq t \leq t_0 + s$, we have $t = T - t \geq 1$ from our assumption, therefore from Lemma I.1:

$$\begin{aligned}
\frac{d}{dt}\|y_t - \tilde{y}_t\| &= \frac{\langle y_t - \tilde{y}_t, \nabla \log q_t(y_t) - \nabla \log q_t(\tilde{y}_t)\rangle}{\|y_t - \tilde{y}_t\|} \\
&= \frac{\int_{r=0}^{1} (y_t - \tilde{y}_t)^\top \nabla\left(y_t + q_t(y_t + r(\tilde{y}_t - y_t))\right)(y_t - \tilde{y}_t)\, dr}{\|y_t - \tilde{y}_t\|} \\
&\leq e^{-t}(2 + 4R^2)\frac{\|y_t - \tilde{y}_t\|^2}{\|y_t - \tilde{y}_t\|} \\
&= e^{-t}(2 + 4R^2)\|y_t - \tilde{y}_t\|,
\end{aligned}$$

which implies

$$\frac{d}{dt}\ln\left(\|y_t - \tilde{y}_t\|\right) \leq e^{-t}(2 + 4R^2). \tag{36}$$

Integrating Equation (36) from $t = t_0$ to $t = t_0 + s$ we get the desired result for the first part. Similarly for the case when $T - t_0 < 1$:

$$\frac{d}{dt}\ln\left(\|y_t - \tilde{y}_t\|\right) \leq \frac{5R^2}{t^2}, \tag{37}$$

which implies (using inequality $\sigma_t^2 = 1 - e^{-t} \leq t$ for $t = T - (t_0 + s)$)

$$\|y_{t_0+s} - \tilde{y}_{t_0+s}\| \leq e^{\frac{5R^2}{T-(t_0+s)}s}\varepsilon \leq e^{\frac{5R^2}{\sigma_t^2}s}\varepsilon. \qquad \square$$

**Lemma I.3.** *(Distributional guarantees for probability flow ODE) Along the backward ODE*

$$\dot{y}_t = F_t^*(y_t) \tag{38}$$

*with $F_t^*(y) = y + \nabla \log q_t(y)$ we have with probability $1 - \varepsilon_1$,*

$$\forall t \in [t_0, t_0 + h], \|y_t\| \le e\left(R + \sqrt{d} + \sqrt{\ln(1/\varepsilon_1)}\right),$$

$$\forall t \in [t_0, t_0 + h], \|F_t^*(y_t)\| \le 6\left(R + \sqrt{d} + \sqrt{\ln(1/\varepsilon_1)}\right),$$

$$\forall s_1, s_2, \|y_{s_1} - y_{s_2}\| \le 6(s_2 - s_1)\left(R + \sqrt{d} + \sqrt{\ln(1/\varepsilon_1)}\right)$$

$$\left\|F_{t_0}^*(y_{t_0}) - F_{t_0+h}^*(y_{t_0+h})\right\| \lesssim \sqrt{d}R(R^2 + \ln(1/\varepsilon)).$$

*Proof.* Recall we can write $y_t = X_t + \sigma_t \xi$ where $F_t^*(y) = y_t + \frac{\mathbb{E}^{t,y}X - y_t}{\sigma_t^2}$. Now for time $t_0$, using $\sigma_{t_0}^2 \ge 0.5$ for $t_0 \ge 1$ as we showed in Lemma I.1, we have

$$\left\|F_{t_0}^*(y_{t_0})\right\| = \mathbb{E}\left\|y_{t_0} + \frac{\mathbb{E}^{t,y_{t_0}}X - y_{t_0}}{\sigma_{t_0}^2}\right\| \tag{39}$$

$$\le \left(1 + \frac{1}{\sigma_{t_0}^2}\right)\|y_{t_0}\| + \frac{1}{\sigma_{t_0}^2}R \tag{40}$$

$$\le \frac{3}{2}\|y_{t_0}\| + \frac{R}{2} \tag{41}$$

Now because $\|\xi\|$ is subgaussian, we get with probability at least $1 - \varepsilon_1$,

$$\|y_{t_0}\| \le R + \sqrt{d} + \sqrt{\ln(1/\varepsilon_1)}.$$

On the other hand, from the definition (38)

$$\frac{d}{ds}\|y_s\| \le \|F_s^*(y_s)\| \le \frac{3}{2}\|y_s\| + \frac{R}{2},$$

so up to time $s$, we get

$$\|y_s\| \le (R + \sqrt{d} + \sqrt{\ln(1/\varepsilon_1)})e^{s\left(R + \sqrt{d} + \sqrt{\ln(1/\varepsilon_1)}\right)},$$

which implies from the fact that $h \le \frac{1}{R + \sqrt{d} + \sqrt{\ln(1/\varepsilon_1)}}$,

$$\|y_s\| \le e\left(R + \sqrt{d} + \sqrt{\ln(1/\varepsilon_1)}\right).$$

The second part follows from (41). For the third part we have

$$\|y_{s_1} - y_{s_2}\| = \left\|\int_{s_1}^{s_2} F_s^*(y_s)ds\right\|$$

$$\le \int_{s_1}^{s_2} \|F_s^*(y_s)\|\, ds$$

$$\le 6(s_2 - s_1)\left(R + \sqrt{d} + \sqrt{\ln(1/\varepsilon_1)}\right).$$

For the last part, using Lemma 3.3:

$$(\mathbb{E}\|\partial_t F_t^*(y_t)\|^p)^{1/p} \lesssim \sqrt{d}R(R + \sqrt{p})^2.$$

Integrating from $t_0$ to $t_0 + h$:

$$\left(\mathbb{E}\left\|F_{t_0}^*(y_{t_0}) - F_{t_0+h}^*(y_{t_0+h})\right\|^p\right)^{1/p} \lesssim h\sqrt{d}R(R + \sqrt{p})^2,$$

which implies with probability at least $\varepsilon_1$ we have

$$\left\|F_{t_0}^*(y_{t_0}) - F_{t_0+h}^*(y_{t_0+h})\right\| \lesssim \sqrt{d}R(R + \sqrt{\ln(1/\varepsilon)})^2.$$

This completes the proof. □

