# OpenReview forum: "High-accuracy and dimension-free sampling with diffusions"
_ICML.cc/2026/Conference — ICML 2026 spotlight_

### Official Review · Reviewer_mgaQ · 2026-03-06

**Soundness:** 4
**Presentation:** 3
**Significance:** 4
**Originality:** 4
**Overall Recommendation:** 5
**Confidence:** 4

**Summary:**

The article proposes a new method for sampling using diffusion models. The new method is based on developing a new solver for the involved ODE, using an interplay between low-degree approximation and the collocation method and the authors prove an iteration complexity that scales polylogarithmically on $\frac{1}{\epsilon}$ in Total variation, and the bound doesn't depend explicitly on the dimension but only throughthe effective radius of the support of the target distribution.
Finally, using some iterations of Underdamped Langevin Monte Carlo they are able to provide guarantees in Wasserstein distance.

**Compliance With Llm Reviewing Policy:**

Affirmed.

**Final Justification:**

This contribution is very strong in originality and significance as it opens new directions for the study of diffusion models.
The authors have addressed all of my concerns and therefore I give a final score of 5.

**Key Questions For Authors:**

1) The assumption of the Lipschitz score seems a bit restrictive. Could tje authors specify exactly the points where such assumption is unavoidable? Could a weaker assumption work, for example in obtaining Wasserstein estimates.

2) Could the authors specify the exact reasons that their approach yields better estimates via the proof section?

3) From the narrative of the submission, the choice of basis $\tilde{\phi}_j$ seems important. Are there other choices, for example picking other kind of Chebychev polynomials?

**Limitations:**

Yes

**Strengths And Weaknesses:**

Soundness: To my understanding the submission is technically sound. I have checked the proofs (not entirely closely) and they seem correct. The assumptions are mostly standard (only assumption 2 is a bit restrictive compared to related literature, but still is reasonable ),
and the proof roadmap seems rigorous.

Presentation: The presentation is very good. The presentation flow is very smooth as the authors present the related literature and their contributions in a satisfying way. There is some room of improvement in the sense of explaining the role of their key novelties to obtain the novel results (see question 2).

Significance: The submission greatly improves the state of the art results, proposing a high-accuracy method for diffusion models.
The results are also novel with respect to the dimension dependence. Solving the ODEs appearing in diffusion models through the collocation method opens a new direction for the study of diffusion models.

Originality: The authors propose a new numerical method for diffusion models. Their analysis has some unique elements that are unique to their method, so I believe that the originality of the submission meets the standards of ICML.

---

> ### Author Rebuttal · Authors · 2026-03-31
>
> We thank the reviewer for their positive score and thoughtful engagement with our work! In particular, we are encouraged by the reviewer’s assessment of the significance of our results and the quality of the presentation.
>
> We respond to their three questions below:
> 1. In fact we prove in this work that the true score for a compactly supported measure convolved with Gaussian noise must be Lipschitz, so our only assumption is that the estimate for the score is likewise Lipschitz. In the absence of such an assumption, we are not aware of any results in the literature that are able to improve upon the baseline of $O(d/\epsilon^2)$ iterations proven in [Benton et al. ‘23] and [Conforti et al. ‘23]
> 2. At a high level, the reason why we are able to obtain a high-accuracy guarantee is that collocation, i.e. Picard iteration, converges exponentially quickly to the true ODE solution. To make Picard iteration algorithmic, we then need to work in a low-degree polynomial basis, thus reducing the proof to establishing the key structural result that the velocity field is approximable by a low-degree polynomial in the time variable. To establish our dimension-free rate, we leverage a crucial algebraic trick in our proof of Lemma 3.1. to remove the term that involves the norm squared $\|y\|^2$, namely the norm of the gaussian convolution of the original distribution variable, which grows with the dimension.
> 3. The only property we use about the particular polynomial basis that we chose is that (1) it consists of low-degree polynomials, and (2) it is “$\gamma$-bounded” in the sense of Definition 2.1. It just happens that it is straightforward to prove $\gamma$-boundedness of Chebyshev polynomials, but we do not use any other properties of these polynomials in our proofs.

---

> > ### Author Rebuttal · Reviewer_mgaQ · 2026-04-01
> >
> > The authors have answered all of my questions.

---

> > > ### Author Response · Authors · 2026-04-01
> > >
> > > Thanks for your reply. Would you consider raising your score since we addressed all your concerns?
> > >
> > > Best,

---

### Official Review · Reviewer_2BL8 · 2026-03-12

**Soundness:** 3
**Presentation:** 3
**Significance:** 3
**Originality:** 3
**Overall Recommendation:** 5
**Confidence:** 3

**Summary:**

## Summary
The paper presents a sampling algorithm for diffusion models that achieves an iteration complexity scaling polylogarithmically with the inverse target accuracy, $\text{polylog}(1/\epsilon)$. This improves upon the polynomial dependence $\text{poly}(1/\epsilon)$ characteristic of standard discretization methods (e.g., Euler-Maruyama). The authors achieve this by adapting high-accuracy ODE solvers—specifically the collocation method via Picard iteration—to the probability flow ODE of the reverse diffusion process. The theoretical feasibility of this approach rests on a structural proof bounding the higher-order time derivatives of the score function. By establishing that these time derivatives are bounded, the authors prove the reverse ODE trajectory can be locally approximated by low-degree polynomials, which allows the sampler to take larger step sizes without accumulating dimension-dependent discretization bias.

**Compliance With Llm Reviewing Policy:**

Affirmed.

**Key Questions For Authors:**

## Questions

To bridge the gap between theory and practice, and to strengthen the submission, I have the following questions and suggestions for the authors:

1. **Latent Space Diffusion Experiments:** Could the algorithm be implemented and tested on latent space diffusion models? Since the Picard approximation evaluates the vector field at multiple polynomial nodes (e.g., Chebyshev nodes), applying it to lower-dimensional latent spaces would serve as an optimal testbed to measure actual computational overhead and iteration efficiency.
2. **End-to-End Image Generation:** Are the authors able to run the proposed ODE sampler on simple image datasets by executing the reverse process in the latent space and passing the output through a standard decoder to generate the final pixels? This would validate whether the theoretical $\text{polylog}(1/\epsilon)$ advantages of the collocation method translate into perceptible improvements in sample quality or generation speed under empirical score estimation errors.

**Limitations:**

yes

**Strengths And Weaknesses:**

## Strengths and Weaknesses

### Strengths
* **Improved Complexity Bounds:** The transition from $\mathcal{O}(\text{poly}(1/\epsilon))$ to $\tilde{\mathcal{O}}(\text{polylog}(1/\epsilon))$ iteration complexity provides a rigorous high-accuracy guarantee for diffusion sampling. Furthermore, the bounds are dimension-free, depending instead on the effective radius of the target distribution.
* **Adaptation of Polynomial ODE Approximations:** The paper successfully translates low-order polynomial approximation techniques for ODEs into the diffusion context. Building upon the foundational framework established by Lee, Song, and Vempala in "Algorithmic Theory of ODEs and Sampling from Well-conditioned Logconcave Densities" (arXiv:1812.06243)—which utilized collocation methods and bounded Taylor truncation errors to achieve $\text{polylog}(1/\epsilon)$ complexity for Hamiltonian Monte Carlo (HMC)—this work rigorously maps these high-accuracy ODE principles onto the probability flow ODE.
* **Novel Structural Bounds on Score Derivatives:** In the log-concave sampling setting of Lee et al. (2018), bounding the derivatives of the HMC vector field was manageable because it relied on Cauchy estimates of an explicit, static potential function. In contrast, the score function in a diffusion model is a dynamic, time-dependent quantity. To resolve this, the authors provide a highly innovative structural proof (Lemmas 3.1 and 3.2) that bounds the time derivatives of the score by relating them to the higher-order moments of the conditional posterior distribution. This mechanism is what guarantees the exponential convergence of the Picard iteration over small time windows.

### Weaknesses
* **Lack of Empirical Validation:** The contribution is entirely theoretical. There are no numerical experiments to demonstrate the practical viability, wall-clock efficiency, or numerical stability of the proposed Picard iteration scheme compared to standard higher-order solvers (e.g., DPM-Solver) in realistic settings.
* **Strong Assumptions:** The theoretical results rely on relatively strong assumptions, notably that the score estimation error exhibits sub-exponential tails (Assumption 2) rather than standard $L_2$ bounds, and that the target distribution is a convolved compact support.

---

> ### Author Rebuttal · Authors · 2026-03-31
>
> We thank the reviewer for their positive score and thoughtful engagement with our work!
>
> Regarding the assumptions, we acknowledge that they are slightly stronger, but in exchange, we obtain a dimension-free rate and exponentially improved bounds on the $\epsilon$ dependence. We leave relaxing these assumptions to future work.
>
> Regarding the empirical validation, we provide some proof-of-concept validation on synthetic Gaussian mixture model data here https://ibb.co/6RL9b3dK which demonstrates that our collocation-based sampler outperforms naive Euler and Heun discretization. We leave more substantial evaluations on latent diffusions to future work as it is somewhat out of scope for our primarily theoretical contribution.

---

> > ### Author Rebuttal · Reviewer_2BL8 · 2026-04-01
> >
> > I have went through the author response and maintaining the same score.

---

### Official Review · Reviewer_7MNH · 2026-03-12

**Soundness:** 3
**Presentation:** 3
**Significance:** 3
**Originality:** 3
**Overall Recommendation:** 5
**Confidence:** 4

**Summary:**

The authors propose a new diffusion sampler that achieves dimension-free complexity and polylogarithmic dependence on error tolerance. By approximating the derivative of the ODE solution via a low-degree polynomial, the algorithm allows for long, stable integration steps without dimension-dependent error growth. Theoretical analysis confirms that the method reaches the target distribution in significantly fewer iterations than prior techniques.

**Compliance With Llm Reviewing Policy:**

Affirmed.

**Final Justification:**

The response clarifies that the technique is novel. Additionally, Assumption 2 is crucial as it enables a dimension-free rate. Consequently, I raise the score for significance and originality.

**Key Questions For Authors:**

### Questions
1. Assumption 2 deviates significantly from standard assumptions in the theory of diffusion model convergence. Could this assumption be relaxed within the current theoretical framework?

2. Would it be feasible to evaluate the proposed method on a simple benchmark dataset, such as a Gaussian mixture model? The experimental analysis should clearly illustrate how the proposed method's performance compares to the Euler-Maruyama scheme, which exhibits only polynomial scaling in $1 / \epsilon$.

**Limitations:**

Yes

**Strengths And Weaknesses:**

### Strengths

The paper proposes a novel solver based on the collocation method for diffusion models, achieving significantly improved iteration complexity over existing methods. Furthermore, the authors carefully track the dependence on the parameters $R$ and $\sigma$.

### Weaknesses

1. While the authors emphasize the importance of calculating high-order time derivatives of the score function as a key feature of their argument, this result is not entirely novel in the literature. More specifically, [1] provides bounds on the time derivative of the score function along the ODE trajectory (as seen in equations 64a and 64b in that paper). Similarly, [2] establishes bounds on higher-order derivative with respect to time for a fixed coordinate variable (see Corollary 3). My concern is whether the proposed approach builds upon or significantly extends these prior findings.

2. The number of basis functions in Algorithm 1 was not explicitly defined in the theorems, raising concerns about its per-iteration computational complexity. The dependence of the polynomial degree $D$ on the precision parameter $\epsilon$ should be explicitly quantified in the theoretical analysis.

References:

[1] Li G. et al. Faster diffusion models via higher-order approximation //arXiv preprint arXiv:2506.24042. – 2025.

[2] Stéphanovitch A. Regularity of the score function in generative models //arXiv preprint arXiv:2506.19559. – 2025.

---

> ### Author Rebuttal · Authors · 2026-03-31
>
> We thank the reviewer for their thoughtful feedback and address the weaknesses and questions raised below:
>
> **Weakness 1**:
> We would like to clarify several points regarding these two references. Regarding [1], they indeed bound the time derivative of the score, but note that they cannot obtain a dimension-free rate. The key step in our proof that achieves this is an algebraic trick in our proof of Lemma 3.1 which removes the term that involves the norm squared $\|y\|^2$, namely the norm of the Gaussian noising of the original distribution variable, which grows with the dimension. We also note that the technique in [1] is fundamentally bottlenecked at *polynomial* acceleration in the dependence on $\epsilon$. The reason is that their analysis requires assuming that the number of iterations is exponential in the quantity “$K$” in their final bound.
> Regarding reference [2], we thank the reviewer for bringing this relevant work to our attention. In a nutshell, the result is similar in spirit to ours but ultimately incomparable as it makes assumptions about the distribution that do not necessarily apply in our setting. However, we will make sure to cite this work in a future version of our paper.
> Finally, we would like to stress that the main result of our paper is not Lemma 3.3 in isolation, but rather our overall algorithmic result which achieves a dimension-free and high-accuracy rate.
>
> **Weakness 2**:
> Our main claim is that the velocity field is $\epsilon$-approximated by a degree $k = O(ln 1/\epsilon)$ polynomial, where $\epsilon \le \epsilon_{\rm err}$ (see Line 1558 in the appendix), and therefore we need $k + 1$ many polynomials. Note that the complexity of the Picard iteration is trivially polynomial in this very small degree quantity, and in any case, in our main theorem, we quantify complexity in terms of the number of evaluations of the score estimate oracle.
>
> **Question 1**: We acknowledge that Assumption 2 is stronger, but in exchange, we obtain a dimension-free rate and exponentially improved bounds on the $\varepsilon$ dependence. We leave relaxing this to future work.
>
> **Question 2**: We have implemented collocation-based sampling and compared it to standard Euler discretization, and Heun discretization, of the probability flow ODE for a mixture of $K$ Gaussians in $K$ dimensions for $K = 16$. We have attached the figure here which show our algorithm’s superior scaling in the high-accuracy regime: https://ibb.co/6RL9b3dK

---

> > ### Author Rebuttal · Reviewer_7MNH · 2026-04-02
> >
> > The paper's weaknesses have been addressed. I now recognize that it offers a novel technique, setting it apart from existing literature. The same holds true for the questions.

---

### Official Review · Reviewer_JQiN · 2026-03-13

**Soundness:** 4
**Presentation:** 3
**Significance:** 3
**Originality:** 4
**Overall Recommendation:** 5
**Confidence:** 5

**Summary:**

This work proposed an accelerated solver for the diffusion ODE flow and obtained a theoretical guarantee scaling poly-logarithmically in the target accuracy. This is based a set of structural assumptions, and improves over prior works significantly.

**Compliance With Llm Reviewing Policy:**

Affirmed.

**Final Justification:**

The authors still haven't provided any test in practice (although it is very easy), but I choose to keep my initial positive score.

**Key Questions For Authors:**

Is it possible to remove any of these extra assumptions? In addition, given that R/sigma is often larger than polynomial of d, it is a little misleading to say dimension-free sampling. Lastly, Section 2.2 is used to provide some intuition, but hard to follow, which I think should be made more clear.

**Limitations:**

Such result is based a set of extra assumptions: (1) R/sigma should be very large in practice (such as distributions lie in disjoint support sets); (2) nearly uniform score estimation error (ignore logarithmic factors).

**Strengths And Weaknesses:**

Strength: For an accelerated solver for the diffusion ODE flow, they establish a sample complexity scaling poly-logarithmically in 1/epsilon. This improves the previous complexity bound significantly.

Weakness: Such result is based a set of extra assumptions: (1) R/sigma should be very large in practice (such as distributions lie in disjoint support sets); (2) nearly uniform score estimation error (ignore logarithmic factors).

---

> ### Author Rebuttal · Authors · 2026-03-31
>
> We thank the reviewer for their positive score and thoughtful engagement with our work!
>
> In high dimensions, it is typical for R/sigma to scale polynomially in the dimension at worst. The selling point of our claim however is that our bound doesn’t need to depend *explicitly* on dimension; our “hard Gaussian mixtures” example from the introduction provides one natural context in which this makes a qualitative difference.
>
> We acknowledge that our score estimation error assumption is a bit stronger than what is standard, and leave relaxing this assumption to future work.

---

> > ### Author Rebuttal · Reviewer_JQiN · 2026-04-02
> >
> > Note that R, sigma and d can all be tested very easily in the real dataset (such as Cifar10, Imagenet) and practical diffusion models (such as ScoreSDE, ADM, EDM). Even without GPUs, one can still calculate these numbers. Hence, “hard Gaussian mixtures” example is not reasonable for me given the above fact. If the authors still don't want to discuss the relation of these parameters in practice, I will decrease my score.

---

> > > ### Author Response · Authors · 2026-04-02
> > >
> > > We thank the reviewer for raising this point. Let us clarify the dependence between $R$, $\sigma$, and $d$ in the specific case of images.
> > >
> > > $\textbf{TL;DR}$: In the case of images, for a fixed perceptual noise threshold $\sigma_0$, we have
> > > $R / \sigma_0 = O(\sqrt{d})$.
> > >
> > >
> > > $\textbf{Value of radius}$. Let $x \sim q_{\mathrm{pre}}$, where $x \in \mathbb{R}^d$ is an image and $q_{\mathrm{pre}}$ is a distribution over images. After normalizing pixel values to $[0,1]$, we have $x \in [0,1]^d$, so $|x| \le \sqrt{d}$. In particular, $q_{\mathrm{pre}}$ is supported in a ball of radius at most $R \le \sqrt{d}$.
> > >
> > > $\textbf{Value of noise}$. Our algorithm samples from
> > > $q = q_{\mathrm{pre}} * \mathcal{N}(0,\sigma^2 I_d)$,
> > > i.e., it generates samples of the form
> > > $y = x + \varepsilon$, where $\varepsilon \sim \mathcal{N}(0,\sigma^2 I_d)$.
> > >
> > > Since $x$ represents the clean image, $\sigma$ should be chosen as the largest value that does not noticeably degrade image quality. We denote this value by $\sigma_0$. There are two complementary ways to calibrate $\sigma_0$:
> > >
> > > $\textit{Empirical calibration}$.
> > > One can directly evaluate the effect of Gaussian noise on images. On standard natural images from scikit-image (with pixels normalized to $[0,1]$), we observe that $\sigma_0 \approx 0.03$ corresponds to noise levels that do not noticeably degrade image quality (see our simulations: https://ibb.co/21r42dGt
> > > , https://ibb.co/cGSHYc3
> > > ).
> > >
> > > $\textit{SNR-based calibration}$.
> > > In image processing, it is standard to require the signal-to-noise ratio
> > > $\mathrm{SNR} = \mathbb{E}|x|^2 / \mathbb{E}|\varepsilon|^2$
> > > to be above a threshold $\tau$ for noise to be imperceptible. Since $\mathbb{E}|\varepsilon|^2 = d \sigma^2$ and for natural images $\mathbb{E}|x|^2 \propto d$, the dimension cancels, yielding
> > > $\mathrm{SNR} \asymp 1 / \sigma^2$.
> > >
> > > Thus, requiring $\mathrm{SNR} \ge \tau$ is equivalent to
> > > $\sigma \le \sigma_0 \asymp \tau^{-1/2}$.
> > > Typical perceptual thresholds correspond to $\mathrm{SNR} \approx 10^3$, which gives $\sigma_0 \approx 0.03$, consistent (up to constants) with the empirical value above.
> > >
> > > In particular, $\sigma_0$ is determined by perceptual considerations (i.e., a minimum acceptable SNR) and does not depend on the dimension $d$. Combining this with $R \le \sqrt{d}$ yields
> > > $R / \sigma_0 = O(\sqrt{d})$.
> > >
> > > $\textbf{Conclusion.}$ We hope this clarifies the practical relationship between $R$, $\sigma$, and $d$ in the image setting. That said, we still believe the fact that our rate does not explicitly depend on the dimension is of mathematical interest, in addition to our high-accuracy guarantee in $log(1/\varepsilon)$

---

### Decision · Program_Chairs · 2026-04-30

**Decision:**

Accept (spotlight)

**Comment:**

This is a strong theory paper that provides a new solver for diffusion models with significantly improved complexity scaling. In my opinion this paper will be of broad interest to the community and I nominate it for an oral presentation.